# The sleep-wake distribution contributes to the peripheral rhythms in PERIOD-2

Marieke MB Hoekstra[†‡], Maxime Jan[†], Georgia Katsioudi, Yann Emmenegger, Paul Franken*

Center for Integrative Genomics, University of Lausanne, Lausanne, Switzerland

**Abstract** In the mouse, *Period-2* (*Per2*) expression in tissues peripheral to the suprachiasmatic nuclei (SCN) increases during sleep deprivation and at times of the day when animals are predominantly awake spontaneously, suggesting that the circadian sleep-wake distribution directly contributes to the daily rhythms in *Per2*. We found support for this hypothesis by recording sleep-wake state alongside PER2 bioluminescence in freely behaving mice, demonstrating that PER2 bioluminescence increases during spontaneous waking and decreases during sleep. The temporary reinstatement of PER2-bioluminescence rhythmicity in behaviorally arrhythmic SCN-lesioned mice submitted to daily recurring sleep deprivations substantiates our hypothesis. Mathematical modeling revealed that PER2 dynamics can be described by a damped harmonic oscillator driven by two forces: a sleep-wake-dependent force and an SCN-independent circadian force. Our work underscores the notion that in peripheral tissues the clock gene circuitry integrates sleep-wake information and could thereby contribute to behavioral adaptability to respond to homeostatic requirements.

## Editor's evaluation

This work contributes interesting data to both the circadian and sleep fields as it presents evidence that clock gene expression in peripheral tissues can be regulated in sleep-wake-state- and peripheral-circadian-dependent manners. To support this idea, the authors monitor sleep-wake state, as well as PER2 expression (utilizing a PER2-luciferase system), in both intact or SCN-lesioned freely behaving mice. Analysis of central and peripheral PER2LUC levels, under diverse sleep protocols, and aided by mathematical models allows them to support the idea that in peripheral tissues the clock gene circuitry integrates sleep-wake information, potentially contributing to behavioral adaptability to homeostatically respond to different challenges.

**\*For correspondence:**
paul.franken@unil.ch

[†]These authors contributed equally to this work

**Present address:** [‡]UK Dementia Research Institute at Imperial College London, Division of Brain Sciences, London, United Kingdom

**Competing interest:** The authors declare that no competing interests exist.

## Introduction

The sleep-wake distribution is coordinated by the interaction of a circadian and a homeostatic process (***Daan et al., 1984***). The biological substrates underlying the circadian process are relatively well understood: circadian rhythms in overt behavior of mammals are generated by the suprachiasmatic nuclei (SCN) located in the hypothalamus (***Hastings et al., 2018***). At the molecular level, so-called 'clock genes' interact through negative transcriptional/translational feedback loops (TTFLs), where the CLOCK/NPAS2:ARNTL (BMAL1) heterodimers drive the transcription of their target genes, among them the *Period* (*Per1,2*) and *Cryptochrome* (*Cry1,2*) genes. Subsequently, PER and CRY proteins assemble into repressor complexes that inhibit CLOCK/NPAS2:ARNTL-mediated transcription, including their own. The resulting reduction of the repressor complex allows a new cycle to start. This feedback loop, present in almost each cell of the mammalian body, interacts with other molecular pathways, together ensuring a period of ~24 hr (***Hastings et al., 2018***). The SCN synchronizes

**eLife digest** Circadian rhythms are daily cycles in behavior and physiology which repeat approximately every 24 hours. The master regulator of these rhythms is located in a small part of the brain called the supra-chiasmatic nucleus. This brain structure regulates the timing of sleep and wakefulness and is also thought to control the daily rhythms of cells throughout the body on a molecular level. It does this by synchronizing the activity of a set of genes called clock genes.

Under normal conditions, the levels of proteins coded for by clock genes change throughout the day following a rhythm that matches sleep-wake patterns. However, keeping animals and humans awake at their preferred sleeping times affects the protein levels of clock genes in many tissues of the body. This suggests that, in addition to the supra-chiasmatic nucleus, sleep-wake cycles may also influence clock-gene rhythms throughout the body.

To test this theory, Hoekstra, Jan et al. measured the levels of PERIOD-2, a protein coded for by the clock gene *Period-2*, while tracking sleep-wake states in mice. They did this by imaging a bioluminescent version of the PERIOD-2 protein in the brain and the kidneys, at the same time as they recorded the brain activity, movement and muscle response of animals. Results showed that PERIOD-2 increased on waking and decreased when mice fell asleep. Additionally, in mice lacking a circadian rhythm in sleep-wake behavior – whose changes in PERIOD-2 levels with respect to time were greatly reduced – imposing a regular sleep-wake cycle restored normal PERIOD-2 rhythmicity.

Next, Hoekstra, Jan et al. developed a mathematical model to understand how sleep-wake cycles together with circadian rhythms affect clock-gene activity in the brain and kidneys. Computer simulations suggested that sleep-wake cycles and circadian factors act as forces of comparable strength driving clock-gene dynamics. Both need to act in concert to keep clock-genes rhythmic. The model also predicted the large and immediate effects of sleep deprivation on PERIOD-2 levels, giving further credence to the idea that waking accelerated clock-gene rhythms while sleeping slowed them down. Modelling also suggested that having regular clock-gene rhythms protects against sleep disturbances.

In summary, this work shows how sleep patterns contribute to the daily rhythms in clock genes in the brain and body. The findings support the idea that well-timed sleep-wake schedules could help people to adjust to new time zones. It might also be useful to inform other strategies to reduce the health impacts of shift work.

peripheral clock gene expression rhythms through its rhythmic behavioral, electrical, and humoral output generated across the day (*Schibler et al., 2015*).

Accumulating evidence suggests that, perhaps surprisingly, clock genes are also involved in the homeostatic aspect of sleep regulation (*Franken, 2013*). This is illustrated by the sleep deprivation (SD)-induced increase in the expression of the clock gene *Per2* in tissues peripheral to the SCN, including the cerebral cortex, liver, and kidney (*Curie et al., 2013*; *Curie et al., 2015*; *Franken et al., 2007*; *Maret et al., 2007*). Moreover, the highest level of peripheral *Per2* expression is reached after the time of day mice were awake most, suggesting that also during spontaneous periods of waking *Per2* expression accumulates. Accordingly, lesioning of the SCN, which eliminates the circadian sleep-wake distribution, attenuates the circadian amplitude of clock gene transcripts and proteins in peripheral tissues (*Akhtar et al., 2002*; *Curie et al., 2015*; *Tahara et al., 2012*; *Saini et al., 2013*; *Sinturel et al., 2021*). Together, these studies suggest that sleeping and waking are important contributors to clock gene expression, but dissecting the contribution of the sleep-wake distribution and circadian time is challenging because the two change in parallel.

By simultaneously recording electroencephalogram (EEG), electromyogram (EMG), locomotor activity (LMA), and PER2-dependent bioluminescence signals from cortex and kidney in freely behaving mice, we established that the circadian sleep-wake distribution importantly contributes to the daily rhythmic changes in central and peripheral PER2 levels. To further test this hypothesis, we predicted that (i) in behaviorally arrhythmic SCN-lesioned (SCNx) mice, daily recurring SDs mimicking a circadian sleep-wake distribution will temporarily reinstate high-amplitude PER2 bioluminescence rhythms, and (ii) in intact rhythmic animals, reducing the amplitude of the circadian sleep-wake distribution will result in a reduced amplitude of PER2 rhythms. While daily SDs indeed enhanced the amplitude of peripheral PER2 rhythms in SCNx mice, the protocol used to reduce the amplitude of

the sleep-wake distribution did not reduce PER2 amplitude in all mice. To reconcile the sleep-wake-driven and circadian aspects of PER2 dynamics, we implemented a mathematical model in which waking represents a force that sets in motion a harmonic oscillator describing PER2 dynamics and found that the sleep-wake distribution, also under undisturbed conditions, is an important contributor to the daily changes in PER2 bioluminescence. Moreover, we discovered a second, SCN and sleep-wake independent force with a circadian period that underlay the residual circadian PER2 rhythms in SCNx mice, and that the phase relationship between these two forces is important for predicting the amplitude response in PER2 rhythms to sleep-wake perturbations.

## Results

To quantify PER2 levels, we used mice expressing a knock-in (KI) construct encoding a fused PER2::LUCIFERASE (PER2::LUC) protein and in which changes in emitted bioluminescence can be used as proxy for changes in PER2 protein levels (*Yoo et al., 2004*). *Per2^Luc* KI mice have been used to follow clock gene expression in vivo (*Curie et al., 2015*; *Ohnishi et al., 2014*; *Tahara et al., 2012*; *van der Vinne et al., 2018*). However, in these studies, mice had to be anesthetized for each measurement, while in the set-up used in our study (RT-Biolumicorder; *Saini et al., 2013*), we assessed PER2 bioluminescence continuously in freely moving mice, in central and peripheral tissues. For the central quantification of bioluminescence, we used mice in which the *Per^Luc* construct was back-crossed onto a C57BL/6J (B6) background (see Materials and methods). For the experiments that quantified bioluminescence in the periphery, we used hairless SKH1 mice carrying the *Per2^Luc* KI construct because lack of fur allows for the unobstructed measurement of emitted photons (see also *Figure 1—figure supplement 1* for experimental design and *Figure 1—figure supplement 2A* for imaging of bioluminescence in SKH1 mice). Under standard LD12:12 conditions, SKH1 mice exhibited sleep-wake patterns characteristic of mice, that is, during the light phase they spent more time in both non-rapid eye movement (NREM) sleep and REM sleep relative to the dark phase, the latter being their habitual active phase. Moreover, they showed the typical sleep homeostatic response to a 6 hr SD during the first 18 hr of recovery, both in sleep time and EEG delta power, although the increase in REM sleep did not reach significance levels (*Figure 1—figure supplement 3*).

In three pilot experiments, we optimized our experimental set-up. We established that the most important source contributing to the peripheral bioluminescence signals in the SKH1 mice are the kidneys (*Figure 1—figure supplement 2A*). We have previously shown that the central bioluminescence signal obtained in B6 mice is of cortical origin (*Curie et al., 2015*). To accomplish this, luciferin was infused directly into the brain, the skull locally thinned and equipped with a glass cone, and animals were not shaved, thereby preventing passage of photons from the periphery (see Materials and methods). We confirmed that in B6 mice thus prepared only photons emitted by the brain could be detected (*Figure 1—figure supplement 2A*). Finally, by using mice expressing luciferase under the control of synthetic CAG promotor (CAG-Luc mice; *Cao et al., 2004*), we determined that its substrate luciferin is best delivered through implantable osmotic mini-pumps compared to administration through the drinking water. Under the latter condition, strong daily rhythms in bioluminescence were observed, likely as a result of rhythms in drinking behavior, thereby driving luciferin availability (*Figure 1—figure supplement 2B*).

### Sleep-wake state affects PER2 bioluminescence

It now has been well documented that enforced wakefulness affects *Per2* mRNA and protein levels in various tissues and mammalian species (*Franken, 2013*; *Hoekstra et al., 2019*; *Möller-Levet et al., 2013*; *Vassalli and Franken, 2017*), but it is not known whether circadian rhythms in spontaneous sleep-wake behavior contribute to the daily changes in PER2 levels. To address this question, we equipped *Per2^Luc* KI B6 and SKH1 mice (n = 6 and 5, respectively) with wireless EEG/EMG recorders (i.e., NeuroLoggers) and monitored simultaneously sleep-wake state, PER2 bioluminescence, and LMA under constant darkness (DD; see *Figure 1—figure supplement 1*). Time asleep in mice kept in the RT-Biolumicorder quantified with the NeuroLoggers under DD was similar to that quantified under LD conditions using a tethered EEG acquisition system (48 hr baseline in DD vs. LD in C57BL/6J mice: NREM sleep: 42.2% ± 1.6%, REM sleep 6.1% ± 0.3% of total recording time; n = 6; compared to 41.3 ± 1.1 and 5.2% ± 0.2%, respectively; n = 12, data taken from *Diessler et al., 2018*). Sleep

and EEG have not been recorded in SKH1 mice previously. Also in these mice, similar sleep durations were obtained under the two conditions (48 hr baseline in DD vs. LD: NREM sleep: 40.4% ± 2.1%, REM sleep 5.8% ± 0.9%; n = 5; compared to 40.0% ± 1.0% and 6.9% ± 0.2%, respectively; n = 8; see *Figure 1—figure supplement 3*).

Next, we reproduced the circadian changes in PER2 bioluminescence as well as the response to a 6 hr SD (*Figure 1A*) as described previously (*Curie et al., 2015*) in both peripheral bioluminescence, mainly of renal origin, and central bioluminescence, emitted by the cortex (see Materials and methods for explanation of the SD procedure). Similar to what was observed in that publication, SD elicited a tissue-specific response, with an immediate increase in central PER2 bioluminescence after SD, while in the periphery the response was delayed and PER2-bioluminescence increases were observed in the second and fifth hour of recovery (*Figure 1A*). In addition to these direct effects on PER2 bioluminescence within the first 6 hr of recovery, the SD also caused a long-term reduction of rhythm amplitude during both recovery days; that is, the first (REC1) and last (REC2) 24 hr period of recovery. In the periphery, this decrease amounted to ca. 30% on both days (REC1: 17.5–41.1%; REC2: 18.7–42.4%, 95% confidence intervals [95% CI]). In the central recordings, PER2-bioluminescence amplitude decreased significantly during REC1 (32% [10.2–54.5%]), whereas the decrease during REC2 (22% [0.06–44.8%]) no longer reached significance levels (linear mixed model with fixed conditional effect ['BSL,' 'REC1,' 'REC2'] and random intercept effect ['Mouse']; periphery: BSL vs. REC1 p=0.0014; vs. REC2 p=0.0011; central: BSL vs. REC1 p=0.022; vs. REC2 p=0.082; sinewave-fitted baseline amplitudes: periphery 0.258 [0.15–0.36]; central 0.193 [0.08–0.31 a.u.]). This long-term reduction of PER2-bioluminescence is reminiscent of the long-term SD effects on rhythm amplitude we observed for *Per2* expression and for other clock genes in the cortex (*Hor et al., 2019*).

Although a circadian modulation of PER2 bioluminescence of both peripheral and central origin is evident (see *Figure 1B* and *Figure 1—figure supplement 4*), we observed additional changes in bioluminescence that occurred simultaneously with changes in sleep-wake state, indicating that PER2 bioluminescence increases during wake-dominated periods and decreases during sleep-dominated periods in both tissues. To quantify this observation, transitions from sleep (irrespective of sleep state) to wake and from wake to sleep were selected (see Materials and methods for selection criteria). Examples of the selected transitions are indicated in *Figure 1B* as a hypnogram. A similar number of sleep-to-wake and wake-to-sleep transitions passed selection criteria during the two-and-a-half baseline days (periphery: 31.2 ± 3.9 and 29.0 ± 3.9; central: 25.6 ± 1.4 and 24.6 ± 1.4, respectively; mean ± SEM). Transitions obtained in mice in which central bioluminescence was recorded were shorter (longest common wake period after sleep-to-wake transitions: 39 vs. 57 min; longest common sleep period after wake-to-sleep transitions: 63 vs. 75 min, for central and periphery, respectively). Although PER2 bioluminescence increased during wakefulness and decreased during sleep in both tissues, tissue-specific differences were observed (*Figure 1C*). At sleep-to-wake transitions, tissue differences concerned an initial decrease in peripheral PER2 bioluminescence after wake onset, followed by a steep increase saturating at 125%. In contrast, central levels of PER2 bioluminescence increased from the start and followed a linear time course throughout the waking period. Despite these different dynamics, similar bioluminescence levels were reached in both tissues. After wake-to-sleep transitions, peripheral PER2 bioluminescence initially increased before quickly decreasing and then leveling out at around 84% (*Figure 1C*). In contrast, the central signal decreased linearly throughout the sleep period reaching levels of 91% at the end.

Changes in sleep-wake state are accompanied by physiological alterations, such as changes in body and brain temperature (*Sela et al., 2020*; *Vishwakarma et al., 2021*), which could influence bioluminescence by changing substrate levels and/or the rate of the enzymatic reaction. Although the circadian rhythms of subcutaneous temperature and PER2 bioluminescence are ca. 4 hr out of phase (*Figure 1—figure supplement 6*), this does not exclude the possibility that fast changes in physiology associated with sleep-wake transitions contribute to luciferin availability. Therefore, we assessed sleep-wake-related changes in mice carrying two other luciferase reporter constructs. Besides the CAG-Luc mice we used to decide on the route of luciferin administration (*Figure 1—figure supplement 2B*), we also had access to *Pkg1-Luc* mice in which bioluminescence is under the control of the promotor of the housekeeping gene *Pkg1* (see Materials and methods). As expected from a housekeeping gene, bioluminescence in *Pkg1-Luc* mice was relatively constant and did not increase at sleep-wake transitions (*Figure 1—figure supplement 5*). In contrast, in CAG-Luc mice increases at

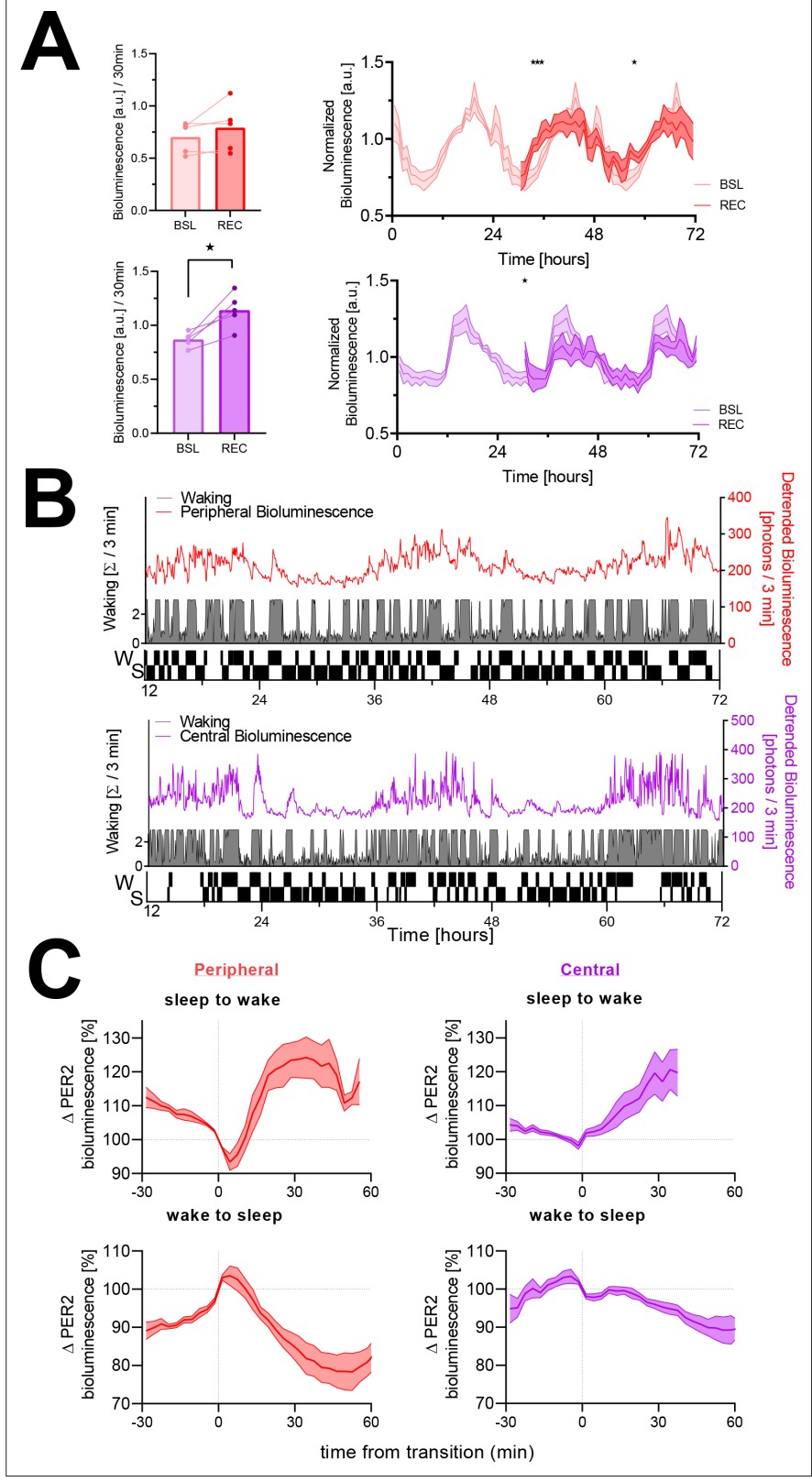

**Figure 1.** The sleep-wake distribution contributes to changes in PER2 bioluminescence. Red, peripheral (kidney) bioluminescence; purple, central (cortical) bioluminescence. (**A**) Left panels: PER2 bioluminescence measured in the first 30 min of recovery (REC) after sleep deprivation compared to levels reached at this circadian time during baseline (BSL). Sleep deprivation elicited an acute response in central PER2 (lower bar graph; t(4) = 4.4, p=0.012),

*Figure 1 continued on next page*

*Figure 1 continued*

but not in the periphery (upper bar graph; t(4) = 1.5, p=0.21). Right panels: under baseline conditions (lighter graphs, time 0–24 hr repeated three times, average of 48 hr baseline), PER2 bioluminescence showed a circadian rhythm both in the periphery (left) and central (right). Sleep deprivation from ZT0 to –6 (times under preceding LD conditions) affected PER2 bioluminescence during recovery (two-way rANOVA w/ Condition × Time, periphery: F(41, 164) = 2.1, p=0.0007; central: F(41, 164) = 1.8, p=0.004). Asterisks indicate significant differences assessed by post-hoc paired *t*-tests. Bioluminescence is expressed as a fraction of the individual average bioluminescence during the experiment and depicted in 1 hr intervals as mean ± SEM for five mice (peripheral and central). (**B**) A sleep-wake state recording (gray area plot represents wakefulness in consecutive 3 min intervals) combined with peripheral (red line, upper graph) and central PER2 bioluminescence (purple line, lower graph) in two mice during baseline. Note that besides the circadian oscillation in PER2 bioluminescence, there are marked increases and decreases in PER2 bioluminescence. The 'hypnogram' (lower part of the graph) illustrates that the rapidly evoked changes in PER2 bioluminescence are related to periods of sleeping (S) and waking (W). This hypnogram is discontinuous as it depicts only the SW transitions selected in this mouse for the analysis in (**C**). (**C**) Changes in PER2 bioluminescence associated with transitions from sleep to wake (top) and wake to sleep (bottom) of peripheral (left, n = 5) and cortical (right, n = 6) origin. Data underlying figures can be found in *Figure 1—source data 1*.

The online version of this article includes the following source data and figure supplement(s) for figure 1:

**Source data 1.** This file contains the numerical values on which the graphs in *Figure 1A–C* are based.

**Figure supplement 1.** Experimental protocol for the three experiments.

**Figure supplement 2.** Sources of bioluminescence and different luciferin administration routes.

**Figure supplement 2—source data 1.** This file contains the numerical values on which the graphs in *Figure 1— figure supplement 2B* are based.

**Figure supplement 3.** Sleep phenotyping of SKH1 mice.

**Figure supplement 3—source data 1.** This file contains the numerical values on which the graphs in *Figure 1— figure supplement 3A–C* are based.

**Figure supplement 4.** PER2 bioluminescence co-detected with EEG-based sleep-wake state in the other mice.

**Figure supplement 4—source data 1.** This file contains the numerical values on which the nine graphs in *Figure 1—figure supplement 4* are based.

**Figure supplement 5.** The effect of activity on bioluminescence in three different bioluminescence reporters.

**Figure supplement 5—source data 1.** This file contains the numerical values on which the graphs in *Figure 1— figure supplement 5A and B* are based.

**Figure supplement 6.** Subcutaneous temperature is phase advanced relative to PER2 kidney bioluminescence.

**Figure supplement 6—source data 1.** This file contains the numerical values on which the graphs in *Figure 1— figure supplement 6A and B* are based.

**Figure supplement 7.** Locomotor activity (LMA) as proxy for wakefulness.

**Figure supplement 7—source data 1.** This file contains the numerical values on which the graphs in *Figure 1— figure supplement 7A and B* are based.

transitions were larger than in *Per2^Luc* mice, consistent with the rapidly activated CMV-immediate-early enhancer contained within the CAG synthetic promotor (*Brightwell et al., 1997*; *Collaco and Geusz, 2003*). These results show that the bioluminescence changes at sleep-wake transitions are specific to the luciferase reporter construct and therefore not an artifact of sleep-wake-related changes in physiology.

Taken together, three observations indicate that the sleep-wake distribution importantly contributes to both central and peripheral PER2-bioluminescence dynamics: (1) enforced wakefulness has both acute and long-term effects on PER2 bioluminescence; (2) the changes in PER2 bioluminescence at sleep-wake transitions demonstrate that also spontaneous waking is associated with an increase, and sleep with a decrease, in PER2 bioluminescence; and (3) that PER2 bioluminescence is high when mice are spontaneously awake more.

The central PER bioluminescence signals tended to be noisier than peripheral signals (signal-to-noise ratio: –1.15 ± 1.0 and 0.37 ± 1.0 dB, respectively; estimated on 3 min values in baseline of the 6 hr SD experiments according to *Leise et al., 2012*). Moreover, the peripheral signal was easier to obtain and minimally invasive to the animal. We therefore decided for the next experiments to focus

on peripheral PER2 bioluminescence using the *Per2^Luc* SKH1 mice. Because waking correlates highly with LMA (*Figure 1—figure supplement 7*), we recorded LMA as proxy for wakefulness to avoid invasive EEG/EMG surgery and mice having to adapt to a ca. 2.7 g recorder mounted on the head.

## Modulating the amplitude of the sleep-wake distribution and its influence on PER2 bioluminescence

The SCN is the main driver of the circadian sleep-wake distribution because animals in which the SCN is lesioned (SCNx) lack a circadian organization of sleep-wake behavior under constant conditions (*Baker et al., 2005*; *Edgar et al., 1993*). Under these conditions, the amplitude of clock gene expression in peripheral organs is significantly reduced, but not eliminated (*Akhtar et al., 2002*; *Curie et al., 2015*; *Tahara et al., 2012*; *Sinturel et al., 2021*). Given our results in freely behaving mice, we expect that imposing a rhythmic sleep-wake distribution in SCNx mice will reinstate high-amplitude rhythmicity in PER2 bioluminescence. Conversely, reducing the amplitude of the circadian sleep-wake distribution in SCN-intact mice is expected to reduce the amplitude of PER2 bioluminescence. We tested these predictions in two complementary experiments (see experimental design in *Figure 1—figure supplement 1*). In the first experiment, we enforced a daily sleep-wake rhythm in arrhythmic SCNx mice by sleep depriving them for 4 hr at 24 hr intervals during four subsequent days. In the second experiment, we aimed to acutely reduce the circadian distribution of sleeping and waking in intact mice according to a '2hOnOff' protocol, comprising 12 2-h SDs each followed by a 2 h sleep opportunity 'window' (SOW), previously utilized in the rat (*Yasenkov and Deboer, 2010*).

## Repeated sleep deprivations in SCNx mice temporarily reinstate a circadian rhythm in PER2 bioluminescence

Lesioning the SCN rendered LMA arrhythmic under DD conditions (*Figure 2—figure supplement 1*). In arrhythmic SCNx mice, we confirmed that rhythms in PER2 bioluminescence were strongly reduced, but not completely abolished (*Figure 2A–C*). During the second baseline measurement after SCNx (BSL2 in *Figure 2B*), we observed in two of the four mice erratic high values (see also *Figure 1—figure supplement 2*) that we cannot readily explain especially because in the subsequent four SD days the variance in the PER2 bioluminescence signal among mice was again as small as in the earlier recordings (*Figure 2A vs. B*). The erratic values in these two mice led to a >10-fold increase in the residual sum of squares (RSS), indicating a poorer fit, and to a higher amplitude of the fitted sinewaves in the second compared to the first baseline recording (*Figure 2C*, BSL1 vs. BSL2). The repeated SDs induced a robust oscillation in the PER2-bioluminescence signal, restoring amplitude to the levels observed prior to SCN lesioning and higher than those observed in the baseline recordings (*Figure 2C*). The latter observation shows that the increased amplitude during the SD did not result from an SD-mediated alignment of different individual phases in baseline. Importantly, the oscillation continued after the end of the SDs while decreasing in amplitude (*Figure 2C*).

## The 2hOnOff protocol reduces the circadian sleep-wake amplitude without consistently modulating PER2 dynamics

In the next experiment, we aimed at reducing the amplitude of the circadian sleep-wake distribution in intact mice using the 2-day 2hOnOff protocol. We measured sleep-wake state and PER2 bioluminescence during baseline conditions, the 2hOnOff procedure, and the two recovery days in two cohorts of mice. In the first cohort, PER2 bioluminescence was monitored, and mice were taken out of the RT-Biolumicorder for the 2 hr SDs. Bioluminescence was therefore quantified only during the SOWs, thus biasing the read-out of PER2 bioluminescence during the 2hOnOff protocol towards levels reached during sleep. A second cohort was implanted with tethered EEG/EMG electrodes to determine the efficacy of the 2hOnOff protocol in reducing the sleep-wake distribution amplitude.

As expected, mice exhibited a circadian PER2-bioluminescence rhythm under baseline conditions (*Figure 3A*). Contrary to expectation, the 2hOnOff protocol did not significantly decrease the ongoing circadian PER2 oscillation when analyzed at the group level (*Figure 3B*). When inspecting the individual responses, four out of the six mice did show a consistent 27% reduction (range 23–31%) in PER2-biolumininescence amplitude during the 2-day 2hOnOff protocol, while in the remaining two mice amplitude increased by 40% (*Figure 3C*). In all six mice, PER2-bioluminescence amplitude reverted to baseline values during recovery, irrespective of whether it was increased or decreased

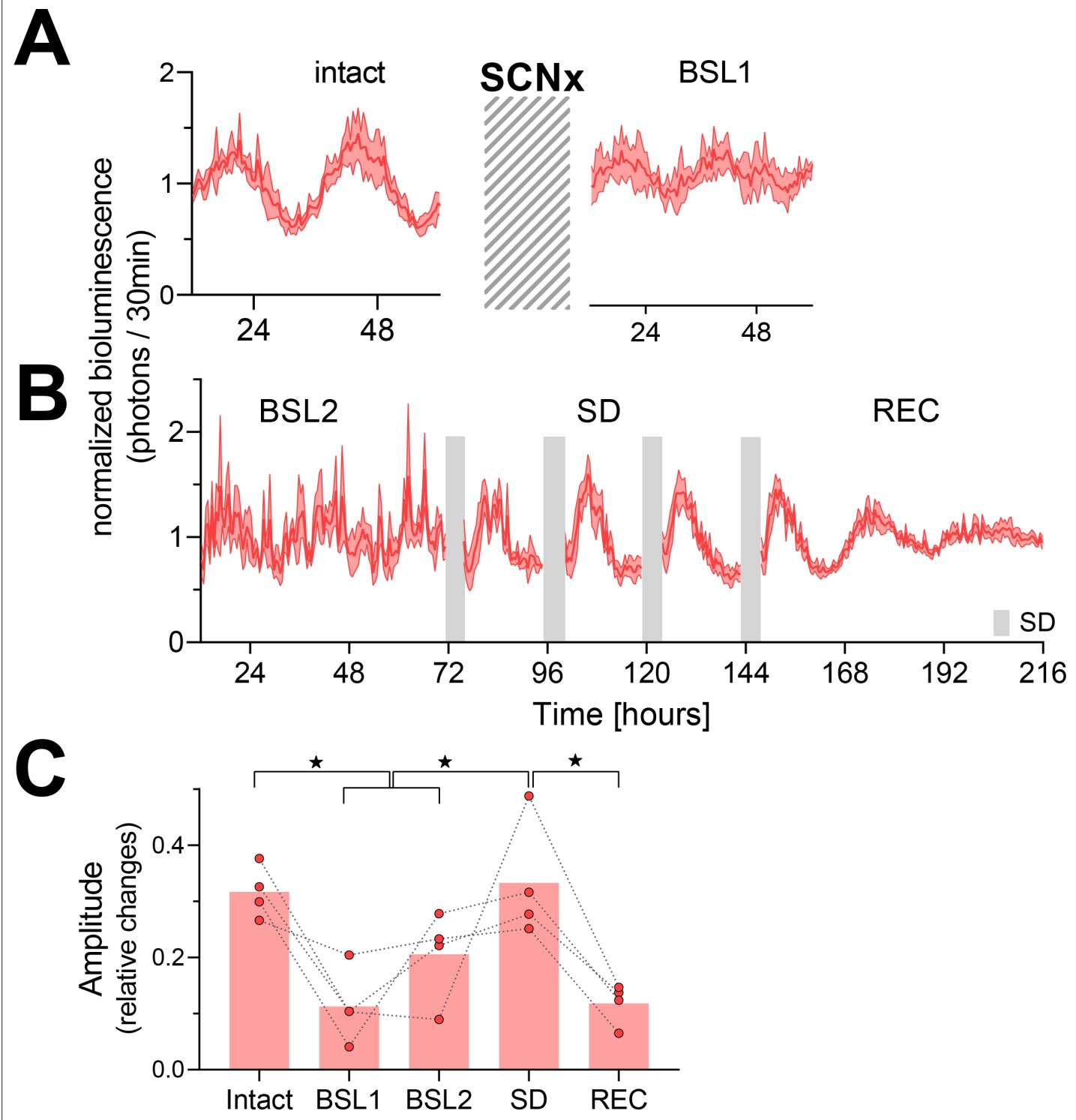

**Figure 2.** Temporary reinstatement of circadian PER2 oscillations in SCN lesion (SCNx) mice by repeated sleep deprivations (SDs). (**A**) Time course of PER2 bioluminescence across 2.5 days under baseline conditions (left) and after the SCNx (right). Cross-hatched area represents an approximately 5-week interval separating the two recordings during which the success of SCNx on locomotor activity was verified under DD (*Figure 2—figure supplement 1*). (**B**) Four 4 h SDs (gray bars) repeated daily at the beginning of the light phase under the preceding LD conditions reinstate rhythmic PER2 bioluminescence (n = 4). Abbreviations of experimental conditions: intact, baseline prior to SCNx; BSL1, baseline after DD locomotor activity recordings; BSL2, baseline immediately preceding the SDs; SD, 20 hr recordings between the four 4 hr SDs; REC, recovery after the four SDs. Data are depicted as mean ± SEM (n = 4). (**C**) Effect of SCNx and SD on PER2-bioluminescence amplitude estimated by sinewave fitting (linear mixed model with fixed conditional effects ['Intact,' 'BSL,' 'SD,' and 'REC'] and random intercept effect ['Mouse'] followed by Tukey's post-hoc tests; intact vs. BSL [BSL1

*Figure 2 continued on next page*

*Figure 2 continued*

and BSL2], p=0.0045; intact vs. REC, p=0.0015; SD vs. BSL: p=0.0013; SD vs. REC: p<0.001; ★p<0.05). See *Figure 2—figure supplement 2* for individual time courses and amplitude estimates of the separate recovery days. Data underlying figures can be found in *Figure 2—source data 1*.

The online version of this article includes the following source data and figure supplement(s) for figure 2:

**Source data 1.** This file contains the numerical values on which the graphs in *Figure 2A–C* are based.

**Figure supplement 1.** Suprachiasmatic nuclei (SCN) lesion eliminates circadian organization of locomotor activity.

**Figure supplement 1—source data 1.** This file contains the numerical values on which the graphs in *Figure 2—figure supplement 1A and B* are based.

**Figure supplement 2.** Individual traces of PER2 kidney bioluminescence.

**Figure supplement 2—source data 1.** This file contains the numerical values on which the graphs in *Figure 2—figure supplement 2A–C* are based.

during the 2hOnOff protocol. The distinct bimodal, opposing response in amplitude observed among individual mice and the subsequent reverting back to baseline suggest that the 2hOnOff protocol did affect the ongoing PER2-bioluminescence rhythm. This phenomenon might relate to factors not accounted for in our experimental design. In the modeling section at the end of the Results section, we explore this observation further and find that variation in circadian phase could underlie the individual differential response (for details, see 'Modeling PER2 dynamics').

We used the second cohort of mice to determine the efficacy with which the 2hOnOff protocol reduced the circadian sleep-wake distribution. In contrast to bioluminescence, the amplitude of the circadian sleep-wake distribution did decrease consistently in all mice (*Figure 3D and E*). The circadian rhythm in the sleep-wake distribution was, however, not eliminated because the sleep obtained during the 2 hr SDs as well as during the 2 hr SOWs both varied as a function of time of day (sleep during SDs: average: 12.4%, min-max: 4.9–23.0%; during SOWs: average: 54.3%, min-max: 35.7–76.4%). This was especially evident at the beginning of the subjective light phase of day 2 when the average time spent asleep during the SD reached 23%. The sleep obtained during the SDs at this time could be due to a substantial sleep pressure because of lost sleep during day 1 of the 2hOnOff protocol (total time spent asleep/day, mean ± SEM; baseline: 11.1 ± 0.1 hr; 2hOnOff: day 1: 7.3 ± 0.3 hr, day 2: 8.6 ± 0.7 hr, paired two-tailed *t*-test, BSL vs. 2hOnOff-day 1: t(7) = 8.86, p<0.001; BSL vs. 2hOnOff-day 2: t(7) = 4.18, p=0.004; 2hOnOff-day 1 vs. -day 2, t(7) = 1.97, p=0.09), combined with the difficulty for the experimenters to visually detect and prevent sleep in pinkish hairless mice under dim-red light conditions. During recovery from the 2hOnOff protocol, baseline amplitudes were re-established (*Figure 3E*).

## Modeling circadian PER2 dynamics

The results above demonstrate that both circadian and sleep-wake-dependent factors need to be considered when studying PER2 dynamics. To understand how these two factors collectively generate the variance in PER2 bioluminescence, we put our experimental data into a theoretical framework and modeled changes in peripheral PER2 bioluminescence according to a driven harmonic oscillator (*Curie et al., 2013*). This type of oscillator is not self-sustained but depends on rhythmic forces to set in motion and maintain the oscillation. In the model, we assume the sleep-wake distribution to represent one such force. In the absence of rhythmic forces, the oscillator loses energy, resulting in a gradual reduction of its amplitude according to the rate set by its damping constant (γ). Besides amplitude and timing (i.e., phase) of the recurring forces and the damping constant, the oscillatory system is further defined by a string constant ($\omega_0^2$) with the natural angular frequency ($\omega_0$) defining the intrinsic period. Because waking was not quantified alongside bioluminescence in the SCNx and the 2hOnOff experiments, we used LMA as a proxy for the driving force provided by wakefulness ($\vec{F}_{WAKE}$). The 6 hr SD experiment showed that LMA and time spent awake are strongly correlated and that their hourly dynamics closely changed in parallel (*Figure 1—figure supplement 7*). Moreover, applying the model to this data set (see below) using either wakefulness or LMA as $\vec{F}_{WAKE}$ yielded similar fits (*Figure 4—figure supplement 1*), demonstrating that LMA is an appropriate proxy for wakefulness for the purpose of our model. We performed the analyses at the group level; that is, the mean LMA level of all mice was used to reflect $\vec{F}_{WAKE}$, and the free parameters in the model were

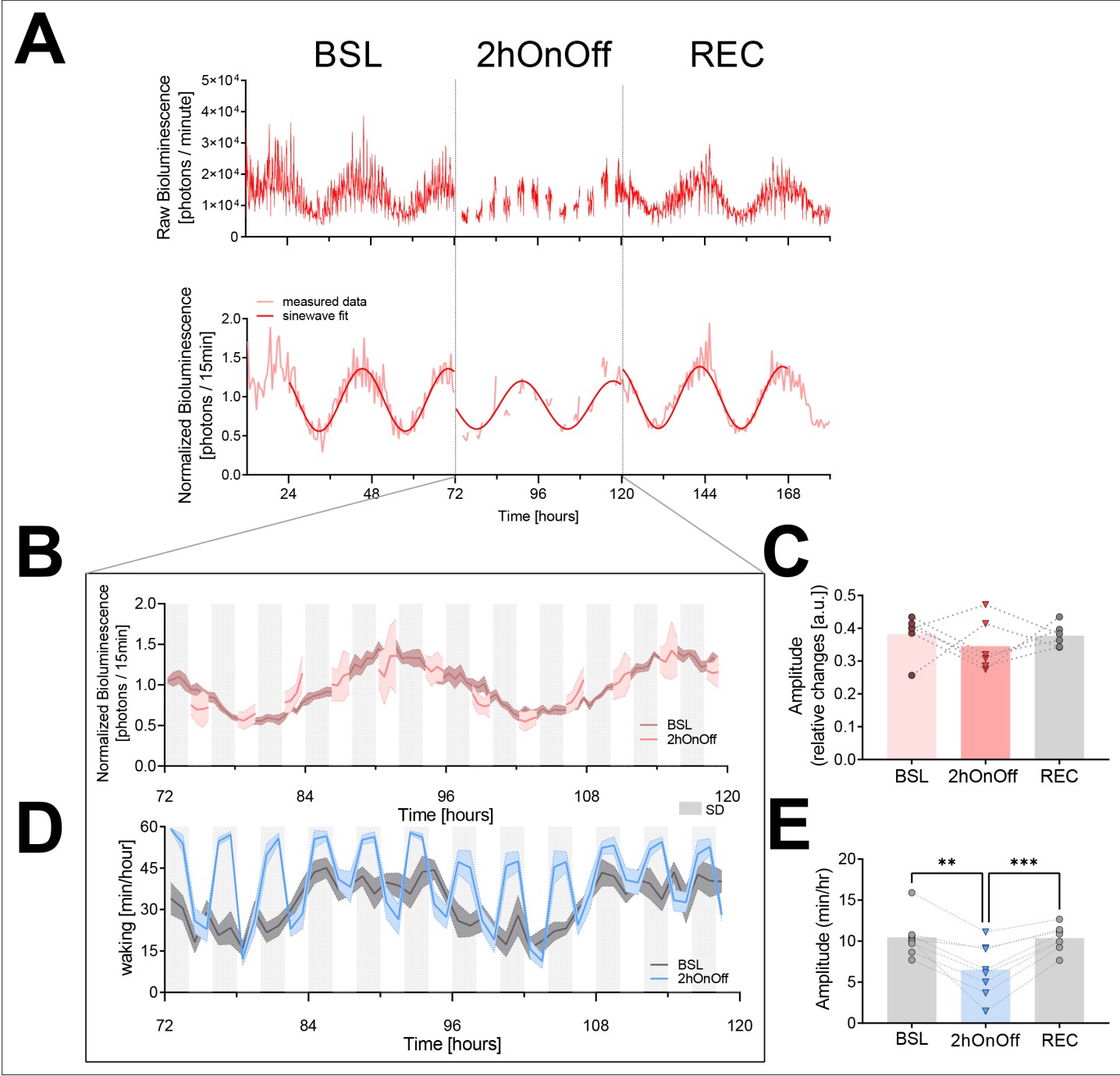

**Figure 3.** The 2hOnOff protocol reduced the circadian sleep-wake amplitude but did not consistently affect PER2 bioluminescence. (**A**) An example of a PER2-bioluminescence recording under baseline, 2hOnOff, and recovery conditions. Upper graph shows bioluminescence data as collected, whereas in the lower graph, the same data are linearly detrended, normalized (relative to the individual overall mean), and averaged over 30 min intervals. A sinewave was fit to the data of the two baseline days, the two 2hOnOff days, and the two recovery days separately (see Materials and methods). (**B**) The time course of PER2 bioluminescence under baseline conditions and during the 2hOnOff protocol (n = 6, data depicted as mean ± SEM). Light gray squares below the graph mark the 2 hr sleep deprivations (SDs). (**C**) The amplitude of the PER2-bioluminescence rhythm decreased in four but increased in two mice, resulting in an overall lack of an effect of the 2hOnOff protocol (paired *t*-test, t(5) = 0.74, p=0.50; mean ± SEM; BSL: 0.38 ± 0.03; 2hOnOff: 0.35 ± 0.03; REC: 0.38 ± 0.01). For all six animals, the amplitude reverted to baseline during recovery. (**D**) The distribution of waking across the two baseline days (dark gray) and during the 2hOnOff protocol (blue) in EEG-implanted SKH1 mice (n = 8, hourly values depicted as mean ± SEM) and a sinewave fit through the hourly average for visual comparison of BSL to SD. (**E**) Individual estimates of the amplitude of circadian changes in wakefulness during the 2hOnOff protocol were obtained by fitting a sinewave function to the wakefulness present in consecutive 4 hr intervals (i.e., SD + SOW). This amplitude was smaller compared to the amplitude obtained in baseline using the same approach (one-way ANOVA, F(1.8, 12.3) = 21.9, p=0.0001; post-

*Figure 3 continued on next page*

*Figure 3 continued*

hoc paired *t*-test, BSL vs. 2hOnOff: t(7) = 4.9, p=0.002; 2hOnOff vs. REC: t(7) = 6.3, p=0.0004; mean ± SEM; BSL: 10.4 ± 0.9; 2hOnOff: 6.5 ± 1.1; REC: 10.4 ± 0.6). However, a circadian modulation was still present under the 2hOnOff protocol (amplitude > 0, one-sample *t*-test, t(7) = 5.8, p=0.0007). Note the overall higher levels of wakefulness during 2hOnOff compared to baseline. Data underlying figures can be found in *Figure 3—source data 1*. BSL, baseline; REC, recovery; SOW, sleep opportunity window.

The online version of this article includes the following source data for figure 3:

**Source data 1.** This file contains the numerical values on which the graphs in *Figure 3A–E* are based.

optimized by fitting the motion of the oscillator to the mean PER2-bioluminescence levels (see Materials and methods). Besides $\overrightarrow{F}_{WAKE}$, we implemented a circadian force to account for the residual PER2-bioluminescence rhythm observed in the SCNx mice (*Figure 2*). We assumed that this additional SCN and sleep-wake-independent force reflects a peripheral circadian process ($\overrightarrow{F}_{PERI}$) present in both intact and SCNx mice (*Sinturel et al., 2021*). This force was modeled as a sinewave with amplitude and phase as free parameters in both experiments while period was set to the period estimated from the baseline PER2-bioluminescence rhythm observed in the SCNx mice (23.7 ± 0.6 hr; n = 4). A schematic overview of the influence of $\overrightarrow{F}_{PERI}$ and $\overrightarrow{F}_{WAKE}$ on PER2 bioluminescence is presented in *Figure 4A*.

We first optimized the parameters of the model describing PER2-bioluminescence dynamics in the SCNx experiment (data from *Figure 2*), and then predicted PER2 bioluminescence under the 2hOnOff experiment. $\overrightarrow{F}_{PERI}$'s amplitude and phase again required optimization as both differed between the two experiments (nonoverlapping 95% CI for both in *Table 1*). With the parameters listed in *Table 1*, the model captured the dynamic changes in PER2 bioluminescence in the SCNx experiment with high precision including the residual rhythmicity in baseline, the reinstated pronounced rhythmicity during the four SDs, and its subsequent dampening thereafter (*Figure 4B*, black line, *Table 2*). The model also accurately captured average bioluminescence dynamics in the 2hOnOff experiment (*Figure 4C*, black line, *Table 2*). It furthermore predicted an 18% reduction of PER2-bioluminescence amplitude during the 2hOnOff protocol compared to baseline (relative amplitudes estimated by the model: 0.31 vs. 0.38 [a.u.]; *Figure 4C*), consistent with our hypotheses and the 27% reduction in PER2-bioluminescence amplitude observed in the four mice in which amplitude did decrease (see *Figure 3C*). To further test the performance of the model, we predicted the peripheral PER2-bioluminescence data obtained in the 6 hr SD experiment using the parameters optimized for the 2hOnOff experiment (*Figure 1A*, *Table 1*). Also, the results of this experiment could be accurately predicted with the model, including the higher PER2-bioluminescence levels reached over the initial recovery hours after SD and the subsequent longer-term reduction in amplitude (*Figures 1A and 4D*, *Table 2*). The model could not reliably predict the central PER2-bioluminescence dynamics mainly due to an earlier phase of the central compared to the peripheral signal (*Figure 4—figure supplement 2D*). Moreover, the increase in PER2 levels immediately following the SD (*Figure 1A*) was missed by the model, further underscoring the tissue-specific relationship between time-spent-awake and PER2 requiring the model to be optimized according to the tissue under study.

The model accurately captured peripheral PER2-bioluminescence dynamics under three different experimental conditions. The model's robust performance prompted us to assess in silico whether (1) both forces are required to predict PER2-bioluminescence dynamics, (2) lesioning the SCN affects the forces exerted on PER2 bioluminescence under undisturbed, DD conditions, and (3) differences in the circadian phase could predict the opposing response in PER2-bioluminescence amplitude among mice in the 2hOnOff experiment.

To evaluate if $\overrightarrow{F}_{PERI}$ and $\overrightarrow{F}_{WAKE}$ combined are required to predict PER2 dynamics, we compared the performance of the model by dropping these two modeling terms; that is, removing either $\overrightarrow{F}_{PERI}$ or $\overrightarrow{F}_{WAKE}$. Removing $\overrightarrow{F}_{PERI}$ from the model did not change the coefficient obtained for $\overrightarrow{F}_{WAKE}$ (8.25e-5, which is still within the 95% CI estimated in the full model; see *Table 1*). However, this simpler model (only four free parameters to estimate compared to six in the full model) could not reliably predict the bioluminescence data of the SCNx and 2hOnOff experiments (*Figure 4B and C*, blue lines) and fit statistics indicated a poorer fit (i.e., a higher Bayesian information criterion [BIC]

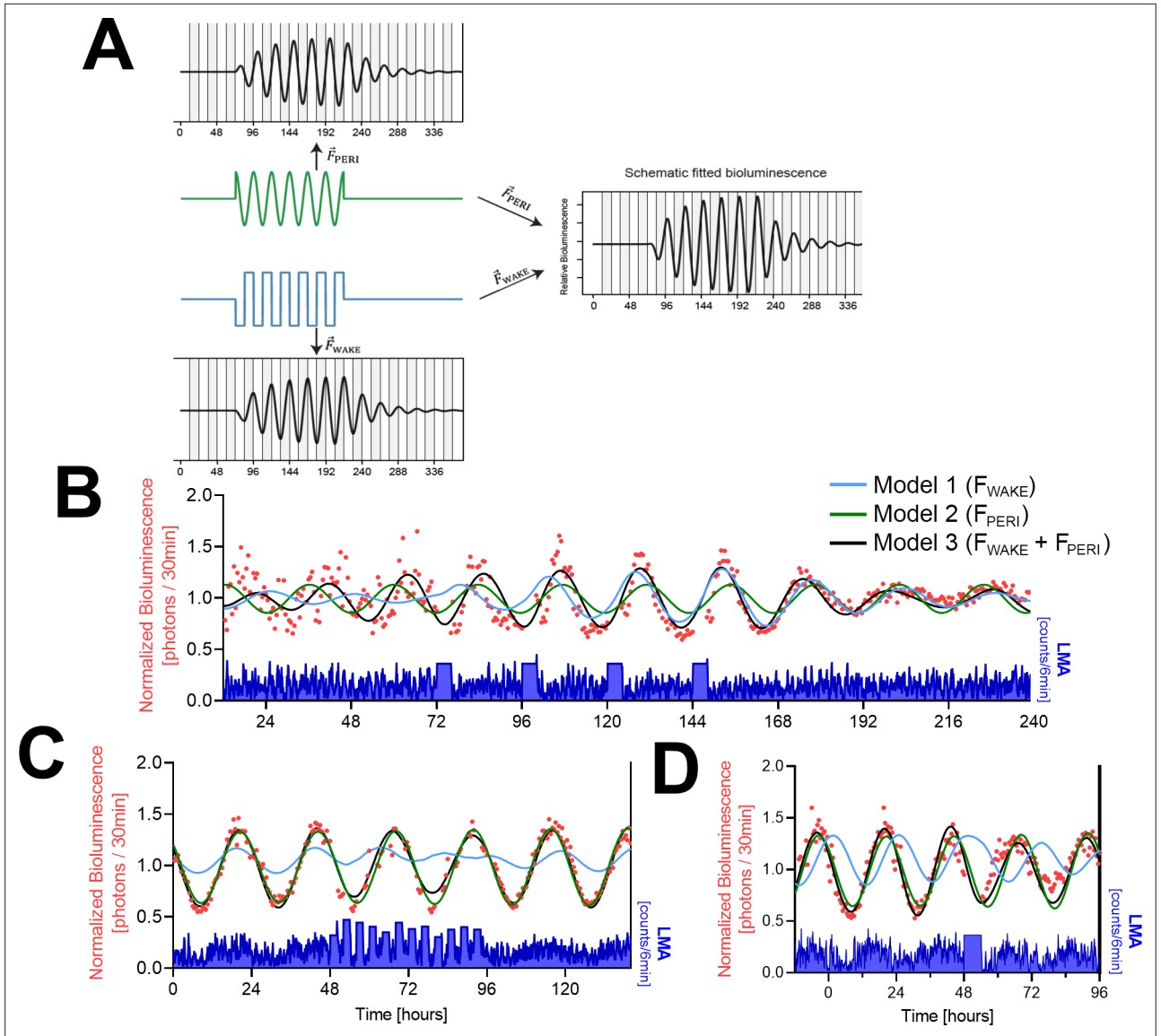

**Figure 4.** Mathematical modeling of PER2-bioluminescence dynamics. (**A**) Schematic view of a driven damped harmonic oscillator. The oscillation (black) is assumed to be driven by two forces: $\vec{F}_{WAKE}$ (blue) and $\vec{F}_{PERI}$ (green). In our model, $\vec{F}_{WAKE}$ is based on locomotor activity (LMA) (here simplified as a square wave) and $\vec{F}_{PERI}$ on a sinewave. Left panels show the individual effect of each force on the oscillator. Both forces start at t = 72 hr and end at t = 216 hr, illustrating the waxing and waning of the resulting rhythm amplitude. Right panel shows the resulting changes in PER2 bioluminescence when combining both forces. Note that combining the two forces increased amplitude and changed the phase of the oscillation. In this example, the amplitude of the peripheral circadian force is flat at beginning and end to illustrate that the oscillation is not self-sustained.

(**B**) Modeling of the SCNx experiment with the full model (model 3; black line) driving oscillations in PER2 bioluminescence or with either $\vec{F}_{WAKE}$ (model 1; blue line) or $\vec{F}_{PERI}$ (model 2; green line) as the only driving force. Red symbols, 30 min PER2-biolumininescence averages. Lower graph in blue: 6 min LMA values. (**C**) Simulation of the PER2-biolumininescence in the 2hOnOff experiment using parameter estimates listed in **Table 1**. Details and legend as in (**B**). (**D**) Simulation of peripheral PER2-biolumininescence in the 6 hr SD experiment using parameter estimates obtained in (**C**) (see **Table 1**). Legend as in (**B**). Data underlying figures can be found in **Figure 4—source data 1**.

The online version of this article includes the following source data and figure supplement(s) for figure 4:

**Source data 1.** This file contains the numerical values on which the graphs in **Figure 4B–D** are based.

**Figure supplement 1.** Locomotor activity (LMA) as proxy for wakefulness.

**Figure supplement 1—source data 1.** This file contains the numerical values on which the graph in **Figure 4—figure supplement 1** is based.

**Figure supplement 2.** Predicting PER2 bioluminescence considering $\vec{F}_{PERI}$ 's phase (**A–C**) and bioluminescence emitted by cortical regions (**D**).

*Figure 4 continued on next page*

*Figure 4 continued*

**Figure supplement 2—source data 1.** This file contains the numerical values on which the graphs in *Figure 4—figure supplement 2C and D* are based.

score, see *Table 2*). For instance, without $\vec{F}_{PERI}$ the model is unable to capture the residual PER2 rhythmicity in the baseline of the SCNx experiment (*Figure 4B*, 'model 1' until 72 hr), and it took longer for the PER2 bioluminescence to reach high-amplitude levels compared to the full model because of $\vec{F}_{PERI}$ being in phase with the timing of the SDs in the full model. With the same strategy, we evaluated the model's performance when dropping $\vec{F}_{WAKE}$ (*Figure 4B and C*, green lines). As the sleep-wake distribution no longer affects the model, $\vec{F}_{PERI}$ dynamics are unperturbed throughout the experiment and can therefore be solved as a sinewave in the differential equation. Fit statistics for the SCNx experiment were poor compared to the full model as the effects of SDs on PER2 amplitude could not be captured (*Figure 4B*, *Table 2*). Removing $\vec{F}_{WAKE}$ when modeling the 2hOnOff experiment resulted in BIC scores similar to those obtained with the full model (*Table 2*), although the amplitude reduction during the 2hOnOff protocol could not be captured (*Figure 4C*). Also, the PER2-bioluminescence dynamics in the 6 hr SD experiment were captured best when using both $\vec{F}_{WAKE}$ and $\vec{F}_{PERI}$ (*Figure 4D*, *Table 2*). Together, these results demonstrate that both forces are required to accurately predict PER2 dynamics under the tested experimental conditions.

To answer the second question, we compared the forces driving PER2 bioluminescence in the baseline recordings of the 2hOnOff (intact mice) and the SCNx experiments. The model estimated that the average absolute force ($\vec{F}_{WAKE}$ , $\vec{F}_{PERI}$ , $\vec{F}\gamma$, and $\vec{F}\omega_0^2$ combined) exerted on the relative PER2-bioluminescense levels was more than twice as high in intact mice compared to SCNx mice (2.11e-2 vs. 0.90e-2 h-2). This large difference was not due to a difference in $\vec{F}_{WAKE}$ , which was somewhat lower in intact mice (2hOnOff vs. SCNx; 2.01e-3 vs. 2.22e-3 h-2). Although $\vec{F}_{PERI}$ was higher by 32% (2.45e-3 vs. 1.85e-3 h-2), this difference did not substantially contribute to the higher amplitude of the PER2 rhythm in intact mice. To illustrate this, we substituted the value of $\vec{F}_{PERI}$ obtained in the 2hOnOff baseline with the lower value obtained in the SCNx baseline, which resulted in a 17% amplitude reduction of the PER2 rhythm. The presence of a circadian sleep-wake distribution impacted

**Table 1.** Parameter estimates obtained in the model optimization for the suprachiasmatic nuclei lesion (SCNx) and 2hOnOff experiments.

Damping constant, natural angular frequency, and the $\vec{F}_{WAKE}$ coefficient were optimized in the SCNx experiment and then used to predict the results of the 2hOnOff experiment. Amplitude and phase of $\vec{F}_{PERI}$ were optimized for both experiments separately. The natural angular frequency defines the intrinsic period of the harmonic oscillation with 0.288 rad * hr$^{-1}$ corresponding to a period of 21.8 hr and its square, $\omega_0^2$, is referred to as the string constant. Phase of the sine function describing $\vec{F}_{PERI}$ is expressed as radians and corresponds to maximum values reached at time 11.8 and 19.4 hr for the SCNx and 2hOnOff experiment, respectively. Values in parenthesis represent the 95% CI.

| Parameters | SCNx | 2hOnOff |
|---|---|---|
| $\gamma$(damping constant) | 0.0155 (0.0103–0.0223) (hr$^{-1}$) | |
| $\omega_0$(natural angular frequency) | 0.288 (0.285–0.291) (rad * hr$^{-1}$) | |
| $\beta$(coefficient $\vec{F}_{WAKE}$) | 6.81e-5 (4.48e-5–9.01e-5) | |
| Model intercept | 0.92 (0.90–0.95) | 0.91 (0.89–0.92) |
| $A$(amplitude $\vec{F}_{PERI}$) | 2.92e-3 (2.58e-3–3.32e-3) | 3.87e-3 (3.80e-3–4.00e-3) |
| $\varphi$(phase $\vec{F}_{PERI}$) | 4.73 (4.6–4.85) | 2.78 (2.78–2.82) |

**Table 2.** Fit statistics for the suprachiasmatic nuclei lesion (SCNx), 2hOnOff, and 6 hr sleep deprivation (SD) experiments using a single force ($\vec{F}_{WAKE}$ or $\vec{F}_{PERI}$) or two forces combined ($\vec{F}_{WAKE}$ and $\vec{F}_{PERI}$).

Bayesian information criterion (BIC) for each model (lower is better; a BIC difference between two competing models larger than 10 is considered strong support for the model with the lower value). Residual sum of squares (RSS) minimized by the model (lower values reflect a better fit). Support of the driven harmonic model compared to a flat model using Bayes factor (>100 is considered a 'decisive' support for the driven harmonic model). Values should be compared only within the same experiment because variance and sample size differed among the experiments.

| | | BIC | RSS | Bayes factor (model vs. flat) |
|---|---|---|---|---|
| | $\vec{F} = \vec{F}_{WAKE}$ | –287.0 | 11.94 | 2.47e32 |
| | $\vec{F} = \vec{F}_{PERI}$ | –224.5 | 13.60 | 6.48e18 |
| SCNx | $\vec{F} = \vec{F}_{WAKE} + \vec{F}_{PERI}$ | –427.5 | 8.36 | 7.87e62 |
| | $\vec{F} = \vec{F}_{WAKE}$ | –28.5 | 11.96 | 1.82e24 |
| | $\vec{F} = \vec{F}_{PERI}$ | –536.8 | 1.29 | 4.41e86 |
| 2hOnOff | $\vec{F} = \vec{F}_{WAKE} + \vec{F}_{PERI}$ | –540.1 | 1.27 | 2.27e87 |
| | $\vec{F} = \vec{F}_{WAKE}$ | 18.2 | 92.50 | 9.15e-17 |
| | $\vec{F} = \vec{F}_{PERI}$ | –205.9 | 4.16 | 5.80e48 |
| 6 hr SD | $\vec{F} = \vec{F}_{WAKE} + \vec{F}_{PERI}$ | –311.9 | 2.46 | 6.23e71 |

PER2 amplitude to a larger extent: running the simulation with the parameters estimated for the 2hOnOff experiment but with the sleep-wake distribution of the SCNx mice resulted in a 32% reduction. Therefore, the circadian sleep-wake organization is an important contributor to high-amplitude oscillations in PER2. In the 2hOnOff experiment, the larger PER2 momentum resulted in a four times higher string and damping forces (2hOnOff vs. SCNx; $\vec{F}\omega_0^2$ : 19.9e-3 vs. 4.4e-3 h-2; $\vec{F}\gamma$: 9.74e-4 vs. 2.52e-4 h-2) that together with the larger $\vec{F}_{PERI}$ underlie the larger average absolute force. Moreover, this analysis demonstrated that the direct effects of $\vec{F}_{WAKE}$ and $\vec{F}_{PERI}$ on PER2 bioluminescence do not depend on an intact SCN and that the magnitude of the two forces is comparable.

Although the model predicted the expected decrease in PER2-bioluminescense rhythm amplitude in the 2hOnOff experiment, this decrease was not observed in the mean bioluminescence data. The individual data showed, however, that in four mice this intervention did decrease PER2-bioluminescence amplitude, while in the remaining two amplitude increased (*Figure 3C*). With the model, we addressed the third question: Do circadian phase differences predict the opposite effects of the 2hOnOff intervention on amplitude? Surprisingly, when systematically varying the phase of $\vec{F}_{PERI}$ during the baseline prior to the 2hOnOff intervention (*Figure 4—figure supplement 2C*), we found that at phase advances larger than 2.5 hr, the model predicted an increase of PER2 amplitude during the subsequent 2hOnOff protocol instead of a decrease (illustrated in *Figure 4—figure supplement 2A* for a 6 hr phase advance). Circadian phase differences among animals might relate to individual differences in period length of their free-running rhythms that accumulate over time. We did not find evidence for a difference in PER2 bioluminescence or LMA phase at the end of the baseline recording (2.5 days under DD) between animals that showed a decrease in PER2-bioluminescence amplitude compared to animals that showed an increase. However, during the first day of recovery following the 2hOnOff protocol (5.5 days under DD), we observed a ca. 2 hr phase advance in bioluminescence

and 1 hr phase advance in LMA in the two mice that increased their amplitude during the preceding 2hOnOff protocol (*Figure 4—figure supplement 2B*, left) compared to the four mice for which we observed the anticipated decrease in PER2 amplitude (*Figure 4—figure supplement 2B*, right). Whether these differences in phase contributed to the opposite response cannot be answered with the current data set. Nevertheless, the model yielded a perhaps counterintuitive but testable hypothesis by demonstrating that phase angle between the sleep-wake distribution and peripheral circadian clock-gene rhythms is an important variable in predicting outcome and emphasizes the importance of carefully controlling the initial conditions.

## Discussion

In this study, we assessed the contribution of the sleep-wake distribution to circadian peripheral PER2 rhythmicity. We presented four key findings, supporting the notion that sleep-wake state is indeed an important factor in determining the circadian amplitude of peripheral changes in PER2: (1) spontaneous sustained bouts of waking and sleep were associated with increased and decreased PER2 bioluminescence, respectively; (2) a single SD acutely increased PER2 bioluminescence and during subsequent recovery dampened its circadian amplitude; (3) repeated SDs temporarily reinstated robust rhythmicity in PER2 bioluminescence in behaviorally arrhythmic mice; and (4) mathematical modeling suggests that PER2 dynamics is best understood as a harmonic oscillator driven by two forces: a sleep-waking-dependent force ($\overrightarrow{F}_{WAKE}$) and an SCN-independent and sleep-wake-independent, circadian peripheral force ($\overrightarrow{F}_{PERI}$).

### How does wakefulness increase PER2?

*Per2* transcription can be initiated from its cognate E-boxes by CLOCK/NPAS2:ARNTL. This transcriptional activation is at the core of the TTFL and drives the circadian changes in PER2. Enforced wakefulness not only affects *Per2* levels but also modulates the expression of other components of the TTLF (*Mang and Franken, 2015*; *Hor et al., 2019*). The SD-evoked increase in *Per2* expression could therefore be mediated through other clock genes in the circuitry, as was demonstrated by the differential SD-evoked response in *Per2* levels in mice lacking the core clock genes *Npas2* and both *Cry1* and *–2* genes (*Franken et al., 2006*; *Wisor et al., 2002*).

Apart from a TTFL-mediated activation, *Per2* transcription can be induced by other signaling molecules directly acting on elements within the *Per2* promotor (*Schibler et al., 2015*). For example, ligand-bound glucocorticoid receptors can induce *Per2* transcription by binding to their glucocorticoid response elements (*Cheon et al., 2013*; *So et al., 2009*). Similarly, cAMP response element (CRE)-binding protein (CREB), heat-shock factor 1 (HSF1), and serum-response factor (SRF) can directly activate *Per2* transcription through CREs, heat-shock elements, and CArG-boxes, respectively, present in the *Per2* gene (*Gerber et al., 2013*; *Saini et al., 2012*; *Tamaru et al., 2011*; *Travnickova-Bendova et al., 2002*). Through these pathways, *Per2* responds to stress, light, temperature, blood-borne systemic cues, and cellular activation as an immediate early gene (IEG). Because of this, *Per2* can appear rhythmic even in the absence of a functional TTFL, provided these signaling pathways fluctuate cyclically (*Kornmann et al., 2007*). In behaviorally arrhythmic SCNx animals, the residual PER2 rhythms we observed might similarly result from SCN-independent corticosterone (*Andrews, 1968*) or body temperature (*Satinoff and Prosser, 1988*) rhythms, or, alternatively, might be TTFL-driven locally (*Sinturel et al., 2021*).

Important for the current study is that several of the pathways known to directly influence *Per2* expression are activated by either spontaneous and/or enforced waking (e.g., corticosterone [*Mongrain et al., 2010*], temperature [*Hoekstra et al., 2019*], Hsf1 and Srf [*Hor et al., 2019*], pCREB [*Cirelli and Tononi, 2000*]) and are therefore good candidates linking sleep-wake state to changes in PER2. The observed changes in PER2 bioluminescence were rapid and suggest that increases in protein can occur within an hour of spontaneous wakefulness. Other studies document that PER genes can indeed be rapidly transcribed and translated. For instance, a light pulse given at CT14 leads within an hour to a significant increase in *Per2* transcript in the SCN (*Yan and Silver, 2002*). One study reported a large increase in hepatic *Per2* transcript levels within 1 hr after food presentation in fasted rats (*Wu et al., 2010*), underscoring the ability of this transcript to rapidly adapt to homeostatic need. *Per2* translation is not solely dependent on de novo transcription and, for example, in the SCN, light

was shown to promote *Per2* translation (*Cao et al., 2015*), suggesting that transcription would not be necessary to increase PER2 protein levels. Such mechanism could also underlie the very fast (<30 min) 1.5-fold increase in PER2 protein observed in fibroblasts after serum shock (*Cao et al., 2015*).

Finally, as the PER2 protein levels measured are the net result of translation and degradation, also sleep-wake-dependent changes in PER2-degradation rate may contribute both to its increase during wakefulness and its decrease during sleep. PER2 degradation is crucial in setting TTFL period and the timing and stability of the circadian sleep-wake distribution (*Chong et al., 2012*; *D'Alessandro et al., 2017*). One established pathway leading to PER2 degradation involves *Casein kinase 1* (*Csnk1*)-mediated phosphorylation (*Eide et al., 2005*) followed by the recruitment of the ubiquitin ligase *β-transducin repeat-containing proteins* (*Btrc*) (*Masuda et al., 2020*; *Ohsaki et al., 2008*; *Reischl et al., 2007*). Other kinases, such as *Salt-inducible kinase 3* (*Sik3*) (*Hayasaka et al., 2017*), and phosphorylation-independent ubiquitin ligases, such as *Transformed mouse 3T3 cell double minute 2* (*Mdm2*) (*Liu et al., 2018*, p. 2), also target PER2 for degradation. Using a modeling approach to estimate the role of *Btrc* in circadian period length, *Reischl et al., 2007* estimated a linear decay rate of 0.18/hr for PER2 degradation, which is not inconsistent with the approximately 0.10/hr net decay rate we observed for the PER2 bioluminescence during sleep. However, the dynamics of PER2 degradation have been assessed in a circadian context exclusively, and effects of sleep-wake state have not been quantified previously.

The obvious next step is to determine which pathway(s) contributes to the wake-driven changes in PER2 protein. We already established that the SD-incurred increase in *Per2* in the forebrain partly depended on glucocorticoids (*Mongrain et al., 2010*). Along those lines, restoration of daily glucocorticoid rhythms in adrenalectomized rats reinstates PER2 rhythms in several extra-SCN brain areas (*Segall and Amir, 2010*). To determine the contribution of the aforementioned wake-driven factors, a genetic screen could be deployed where one-by-one the regulatory elements in the *Per2* promoter are mutated, and the effect of sleep-wake driven *Per2* changes is assessed. This approach has already been taken for the GRE and CRE elements in the *Per2* promoter to test their respective roles in circadian phase resetting and integrating light information (*Cheon et al., 2013*; *So et al., 2009*; *Travnickova-Bendova et al., 2002*).

## Insights from the model

The model accurately captured the main features of peripheral PER2 dynamics observed in all three experiments, thereby giving further credence to the notion that sleep-wake state importantly contributes to the changes in PER2 observed in the periphery. Moreover, it demonstrated that the large amplitude of the circadian PER2 rhythm in intact mice is likely the result of the momentum gained in the harmonic oscillator through the daily recurring sleep-wake distribution. This is conceptually different from a currently accepted scenario, in which direct and indirect outputs from the SCN assure phase coherence of locally generated self-sustained circadian rhythms (*Schibler et al., 2015*). According to this model, loss of amplitude observed at the tissue level in SCNx mice is caused by phase dispersion of the continuing rhythms in individual cells. Among the SCN outputs thought to convey phase coherence are feeding, LMA, and changes in temperature. Because these outputs all require or are associated with the animal being awake, it can be argued that in both models the circadian sleep-wake distribution is key in keeping peripheral organs rhythmic.

The harmonic oscillator model further showed that, although important, the sleep-wake force alone was not sufficient to predict PER2 dynamics. In addition to account for the residual PER2 rhythms observed in undisturbed SCNx mice, the SCN-independent and sleep-wake-independent circadian force greatly improved the performance of the model. The synergistic effect of both forces ($\vec{F}_{PERI}$ and $\vec{F}_{WAKE}$) was needed to explain the rapid response to the SDs observed in the SCNx experiment and also in maintaining robust circadian rhythms during the SDs in the 2hOnOff experiment as illustrated in *Figure 4B and C*, respectively. Furthermore, this synergistic effect greatly depended on the relative phase of the two forces as we could illustrate in silico for the 2hOnOff experiment: a relative subtle change in the phase of $\vec{F}_{PERI}$ might underlie the increase (instead of the predicted decrease) in PER2 amplitude in two of the six mice recorded.

Which pathways set the phase of $\vec{F}_{PERI}$ and whether it is truly independent of $\vec{F}_{WAKE}$, as assumed in the model, our current results cannot answer. In an earlier modeling effort using a similar approach in SCN-intact, light-dark entrained mice, we also required a second force to correctly predict the

phase of the observed rhythm in brain *Per2* expression (*Curie et al., 2013*). In that publication, we based the second force on the pattern of corticosterone production sharply peaking at ZT11 just prior to the light-dark transition. The phase of $\vec{F}_{PERI}$ in the 2hOnOff experiment, which followed a more gradual, sinewave function of which values became positive shortly after ZT11 (extrapolated from the preceding LD cycle), seems consistent with this. In the SCNx experiment, the phase of $\vec{F}_{PERI}$ was positioned ~7 hr earlier with a positive driving force starting at the end of each of the SDs. As SD is accompanied by an increase in corticosterone (*Mongrain et al., 2010*), the phase of $\vec{F}_{PERI}$ could be associated with corticosterone signaling also in the SCNx experiment. Thus in intact mice, the SCN output would dictate the phase of corticosterone production in the adrenals (and thus that of $\vec{F}_{PERI}$), while in SCNx mice the phase of the corticosterone rhythm can be reset by stressors such as SD. As PER2 and *Per2* levels in the SCN seem insensitive to SD (*Curie et al., 2015*; *Zhang et al., 2016*), this could explain why the phase of $\vec{F}_{PERI}$ is maintained in sleep-deprived SCN-intact mice. Moreover, ex vivo experiments demonstrated that the adrenal gland can generate bona fide circadian rhythms in corticosterone release independent of the SCN (*Andrews, 1968*; *Engeland et al., 2018*; *Kofuji et al., 2016*), even though SCNx is generally thought to abolish rhythms in circulating corticosterone levels (*Moore and Eichler, 1972*). While rhythmic corticosterone release represents a plausible candidate contributing to $\vec{F}_{PERI}$ , especially considering its role in synchronizing peripheral clocks (*Balsalobre et al., 2000*; *Cuesta et al., 2015*; *Dickmeis, 2009*; *Le Minh et al., 2001*), our current results cannot rule out other sources underlying or contributing to $\vec{F}_{PERI}$ . Above we argued that also wakefulness (i.e., $\vec{F}_{WAKE}$ in the model) could influence peripheral PER2 through corticosterone signaling, further complicating the issue. Indeed, adrenalectomy was found to reduce (but not abolish) the SD-induced increase in *Per2* expression in the forebrain (*Mongrain et al., 2010*). However, enforced but not spontaneous wakefulness is accompanied by increases in corticosterone and the model could predict PER2 dynamics without having to distinguish between the two types of waking. Therefore, other candidate signals among those listed above must be considered to understand the biological basis of $\vec{F}_{PERI}$ and $\vec{F}_{WAKE}$ .

Model optimization yielded an unexpected short 21.8 hr period for the natural frequency of the PER2 oscillator, which, in addition, differed from the 23.7 hr period we set for $\vec{F}_{PERI}$ . While in the intact mice of the 2hOnOff experiment it is difficult to independently determine $\vec{F}_{PERI}$ 's period, we estimated a 23.7 hr period length for the residual PER2 rhythmicity observed during baseline in SCNx mice, which we assume is driven by $\vec{F}_{PERI}$ . In intact mice kept under constant conditions, SCN output drives behavioral sleep-wake rhythms and synchronizes peripheral clock-gene rhythms forcing the entire system to oscillate at the intrinsic period of the SCN. Similarly, in behaviorally arrhythmic SCNx mice, we assume that the period of the observed residual PER2 rhythm reflects that of the only remaining driver, $\vec{F}_{PERI}$ , as $\vec{F}_{WAKE}$ is no longer rhythmic and direct effects of the SCN are absent. Ex vivo experiments showed that periods vary among tissues and do not depend on whether tissues were obtained from an intact or SCNx mouse (*Cederroth et al., 2019*; *Yoo et al., 2004*), pointing to tissue-specific TTFLs, which we assume to underlie the intrinsic rhythmicity of both $\vec{F}_{PERI}$ and PER2 bioluminescence in our experiments. The difference in period length of $\vec{F}_{PERI}$ and that of the intrinsic PER2 oscillator therefore suggests that $\vec{F}_{PERI}$ is not of renal origin; that is, the tissue that contributed most to the bioluminescence signal we recorded.

## Tissue specificity of the relationship between sleep-wake state and PER2

Our data demonstrated that both central and peripheral PER2-bioluminescence dynamics are affected by sleep-wake state, not only after SD but also after spontaneous periods of wakefulness. Despite this general observation, we found clear tissue-specific differences: the 6 hr SD elicited an immediate increase in the central PER2 signal, while in the periphery this increase occurred several hours later, confirming our earlier findings in brain versus liver and kidney (*Curie et al., 2015*). Tissue specificity was also observed after spontaneous bouts of wakefulness: in the brain, PER2 bioluminescence immediately increased after the animal woke up and continued to do so until the end of the waking bout, whereas in the kidney the increase in PER2 bioluminescence became apparent only 5–10 min

after the transition. Similar differences in PER2 dynamics were observed after falling asleep, albeit in opposite direction. Also, the model suggested a tissue specificity as central dynamics of the 6 hr SD experiment could not be accurately predicted with the parameters optimized on peripheral PER2-bioluminescence data. In its current form, the model describes the global effects of external forces (the sleep-wake distribution and the SCN-independent circadian force) on the behavior of the oscillator, but not their acute effects. Translated into molecular terms, the model only makes predictions on the TTFL aspect of PER2 regulation, not on PER2 as an IEG. Accordingly, the model cannot capture the fast changes at the transitions. Similarly, the short-lasting high levels of PER2 observed in the brain immediately after the 6 hr SD might reflect an IEG response rather than the state of the TTFL oscillator, which would explain why the model could not predict it. As sleep-wake states are brain states, it stands to reason that changes in brain PER2 levels capture more of the acute IEG effects than in the periphery. One could test this hypothesis by quantifying the PER2 response after activating a peripheral tissue, provided this can be achieved without affecting sleep-wake state as well.

## Do changes in bioluminescence reflect changes in PER2 levels?

The method we implemented to quantify PER2 protein levels presents advantages over previous methods used. It enabled us to acquire data at a time resolution needed to link changes in PER2 to sleep-wake state transitions in individual mice. Moreover, because of the within-subject experimental design, there is a substantial reduction in data variability, and, as illustrated with the effects of individual phase on PER2 amplitude in the 2hOnOff experiment, we could assess the presence of individual differences in the initial conditions that might influence experimental outcome. Finally, the number of mice needed for these experiments has been greatly reduced while obtaining better quality data.

A limitation of this method is the assumption that changes in bioluminescence reflect changes in PER2 protein levels. Using western blot, we previously validated that changes in bioluminescence during baseline and after a 6 hr SD indeed reflect changes in PER2 protein (*Curie et al., 2015*). Nevertheless, substrate availability can importantly contribute to the signal as demonstrated in the experiment in which we delivered luciferin in the drinking water. Even the use of osmotic mini-pumps does not guarantee constant delivery as its release rate is temperature dependent (Alzet, manufacturer's notes) and bioluminescence's increase during wakefulness might therefore result from the accompanying increase in temperature during this state. Arguments against a possible temperature effect on bioluminescence changes come from a study in which, using the same osmotic mini-pumps, the expression of two clock genes known to oscillate in anti-phase could be confirmed (*Ono et al., 2015*), which would not be possible if temperature was the main determinant of bioluminescence. In the current data, the circadian rhythm in wakefulness in the CAG-Luc and *Pkg1-Luc* mice was not accompanied by changes in bioluminescence. Moreover, we observed that the circadian rhythms of subcutaneous temperature and PER2 bioluminescence are ca. 4 hr out of phase (*Figure 1—figure supplement 6*), supporting that the large circadian changes in bioluminescence are not driven by changes in luciferin availability. Moreover, the lack of an increase in bioluminescence during wake bouts in *Pkg1-Luc* mice demonstrates that changes in rate-limiting availability of luciferin did not contribute to the fast sleep-wake-evoked changes in PER2.

## Conclusions

In this study, we used a unique combination of methods allowing us to collect high-resolution data of sleep-wake state in conjunction with PER2 levels and found that the sleep-wake distribution profoundly affects PER2 bioluminescence both short and long term. Such behavior-dependent plasticity of the time-keeping machinery in tissues peripheral to the SCN enables the organism to respond to challenges as time-restricted feeding experiments have demonstrated (*Damiola et al., 2000*; *Saini et al., 2013*). Besides its importance in regulating feeding and energy homeostasis (*Bass and Takahashi, 2010*), the clock circuitry also plays a prominent role in sleep homeostasis (*Franken, 2013*). PER2 seems perfectly suited as an integrator of sleep-wake state and circadian time because it is sensitive to a variety of sleep-wake-driven signals. Our model suggests that having a large amplitude rhythm protects from acute disturbances of sleep as observed in the 2hOnOff experiment, while sleep-depriving arrhythmic SCNx mice had immediate and large effects on PER2. These rapid effects could only be achieved through the synergistic effect of a second force that we found to be independent of

the SCN and the sleep-wake distribution. The coordination of the sleep-wake force and this second force in the model was critical in predicting the effects of sleep-wake interventions on PER2. Research on the nature of this second force would therefore be important to facilitate phase resetting and normalize disrupted clock gene rhythms under conditions of jet lag and shift work, complementing strategies aimed at altering the timing of the central pacemaker.

## Materials and methods
### Mice and housing conditions
To measure peripheral PER2-bioluminescence levels, we made use of the *Per2^Luc* KI construct (*Yoo et al., 2004*). The KI construct was originally generated on a C57BL/6J-129 mixed background and subsequently brought onto a C57BL/6J (B6) background by backcrossing for at least 11 generations. These mice were then crossed with outbred SKH1 mice (Crl:SKH1-Hrhr; Charles River) to create hairless *Per2^Luc* KI mice. We used male homozygous *Per2^Luc* KI B6 and hairless heterozygous *Per2^Luc* KI SKH1-B6 hybrid (here referred to as SKH1 mice) mice. Mice were kept under a 12 hr light/12 hr dark cycle with light- and dark onset referred to as Zeitgeber time (ZT)-0 and -12, respectively. Age at time of recording varied between 12 and 24 weeks. Food and water was available ad libitum, and after surgery mice were singly housed. All experiments were approved by the Ethical Committee of the State of Vaud Veterinary Office Switzerland under license VD2743, 3201, and 3402.

### Source of bioluminescence and luciferin's route of administration
Because *Per2^Luc* KI mice express ubiquitously luciferase and the RT-Biolumicorder cannot discriminate between different sources of bioluminescence, we assessed which peripheral organ(s) was/were the major source of bioluminescence. Two male heterozygous *Per2^Luc* SKH1 mice were implanted with an osmotic mini-pump (model 1002; 35 mg/mL luciferin) and 5 days later lightly anesthetized with 2.5% isoflurane and imaged for 60 s (Xenogen IVIS Lumina II) around ZT6. The main source of dorsal bioluminescence overlapped with the expected location of the kidney, whereas ventrally almost no bioluminescence was detected (see *Figure 1—figure supplement 2A*). Most bioluminescence quantified during the experiment is of dorsal origin due to the orientation of the mouse relative to the PMT, suggesting that the kidneys are the main source of peripheral bioluminescence in *Per2^Luc* SKH1 mice. To confirm our previous report that the main source of central bioluminescence was the brain (*Curie et al., 2015*), we imaged one male B6 PER2::LUC mouse at ZT3 again using the IVIS system (*Figure 1—figure supplement 2*). When luciferin is infused centrally and the skull is thinned and a glass cone mounted, all emitted bioluminescence originates from the head of the mouse, with no detectable signal from the periphery. For details on the surgery see 'Surgical procedures and experimental design.'.

In a second pilot experiment, we investigated the optimal route of luciferin administration. Although luciferin administration via drinking water has been used before to measure bioluminescence (*Saini et al., 2013*, *Iwano et al., 2018*, *Hall et al., 2018*, *Sinturel et al., 2021*), we were concerned that this route could limit luciferin availability in a circadian fashion because drinking behavior has a strong circadian rhythm (*Bainier et al., 2017*). To address these concerns, we made use of mice expressing constitutively luciferase under control of the synthetic CAG promotor (*Cao et al., 2004*, Jackson catalog number 008450), thus allowing for testing of circadian fluctuating levels of luciferin. Mice received luciferin via the drinking water or via an osmotic mini-pump and served as their own control. Four male CAG-Luc mice were housed for two subsequent experiments in constant darkness in the RT-Biolumicorder. During the first experiment, 0.5 mg/mL luciferin was dissolved in the drinking water. At the end of this experiment, mice received subcutaneously an osmotic mini-pump (Alzet, model 1002) under light anesthesia (isoflurane; 2–4% mixed with $O_2$) containing 70 mg/mL of luciferin and could recover for 2 days before bioluminescence and activity was monitored for the second experiment in the RT-Biolumicorder.

### Surgical procedures and experimental design
Experimental design of the three main experiments is depicted in *Figure 1—figure supplement 1* with the upper panel (Experiment 1) illustrating the central and peripheral recordings of PER2 bioluminescence alongside EEG/EMG and LMA before, during, and after a 6 hr SD, and the middle and

bottom panels (Experiment 2 and 3) the SCNx and the 2hOnOff experiments, respectively. In the latter two experiments, peripheral PER2 bioluminescence and LMA were recorded.

## Sleep-wake state determination in parallel with PER2 bioluminescence

Mice were implanted with EEG and EMG electrodes under deep ketamine/xylazine anesthesia. Three gold-plated screws (frontal, parietal, and cerebellar) were screwed into the skull over the right cerebral hemisphere, where the cerebellar screw served as a reference for the other two electrodes. Two additional screws were used as anchor screws. For the EMG, a gold wire was inserted into the neck musculature along the back of the skull. For brain delivery of D-luciferin, a cannula (Brain Infusion Kit1, Alzet) was introduced stereotaxically into the right lateral ventricle (1 mm lateral, 0.3 mm posterior to bregma, and 2.2 mm deep) under deep anesthesia (ketamine/xylazine; intraperitoneally, 75 and 10 mg/kg, respectively), and connected to the mini-pump. A depression (diameter 2 mm) was made in (but not through) the skull in a region of the left frontal cortex (approximate coordinates 2 mm lateral to midline, 2 mm anterior to bregma), in which a glass cylinder (length 4.0 mm; diameter 2.0 mm) was positioned and fixed with dental cement. The EMG and three EEG electrodes were subsequently soldered to a connector and cemented to the skull. The cerebellar screw served as a reference for the parietal and frontal screw and the EMG. After the first recovery day, mice were habituated to the weight of the wireless EEG recording system by attaching a dummy of same size and weight to their connector. Two days before habituation to the RT-Biolumicorder, mice were implanted with the osmotic mini-pump (model 1002, Alzet; luciferin 35 mg/mL) under light anesthesia. 8–10 days post-surgery, mice were placed in the RT-Biolumicorder at the end of the light phase (~ZT10-ZT12) for 2 days in LD to habituate to the novel environment. At the end of the second habituation day, the dummy was replaced with a wireless EEG (NeuroLogger, TSE Systems GmbH). After two-and-a-half days of baseline recording in constant darkness, mice were sleep deprived for 6 hr at a time they were expected to rest (ZT0 under LD conditions) by gentle handling as described (*Mang and Franken, 2012*). In short, mice are left undisturbed as long as they do not show signs of sleep. Sleep is prevented by introducing and removing paper tissue, changing the litter, bringing a pipet in the animal's proximity, or gentle tapping of the cage. As opposed to what the term might suggest, mice are not handled. After SD, mice were placed back into the RT-Biolumicorder for the subsequent two recovery days.

## SCNx experiment

Four SKH1 mice were recorded over the course of the experiment and served as their own control. Briefly, their PER2-bioluminescence rhythm was monitored before SCNx (once), under undisturbed conditions post-SCNx (twice), and after the second measure under SCNx conditions, the mice were subjected to the repeated 4 hr SDs.

Bilateral lesion of the two SCNs was performed stereotaxically (Kopf Instruments, 963LS, Miami Lakes, FL) under ketamine/xylazine anesthesia (intraperitoneal injection, 75 and 10 mg/kg, at a volume of 8 mL/kg). Two electrodes (0.3 mm in diameter) were introduced bilaterally at the following coordinates (position of the frontal electrode: anteroposterior using bregma as reference: ± 0.2 mm lateral, + 0.5 mm bregma, depth: –5.9 mm; the second electrode was positioned 0.7 mm posterior to the frontal one). Electrolytic lesions (1 mA, 5 s) were made using a direct current (DC) lesion device (3500, Ugo Basile, Comerio, Italy). After lesion, mice were housed in constant dark (DD) conditions for at least 10 days to verify absence of circadian organization of overt behavior. Activity was quantified using passive infrared sensors (Visonic SPY 4/RTEA, Riverside, CA). ClockLab software (Actimetrics, Wilmette, IL) was used for data acquisition and analyses.

## Surgeries for tethered EEG/EMG recordings

SKH1 mice (n = 8) were implanted with EEG and EMG electrodes as described previously (*Mang and Franken, 2012*) to determine sleep-wake state. The surgery took place under deep xylazine/ketamine anesthesia. Briefly, six gold-plated screws (diameter 1.1 mm) were screwed bilaterally into the skull over the frontal and parietal cortices. Two screws served as EEG electrodes, and the remaining four screws anchored the electrode connector assembly. As EMG electrodes, two gold wires were inserted into the neck musculature. The EEG and EMG electrodes were soldered to a connector and cemented to the skull. Mice recovered from surgery during several days before they were connected to the

recording cables in their home cage for habituation to the cable and their environment, which was at least 6 days prior to the experiment. The habituation to the room and the recovery from the 2-day SD procedure took place under LD 12:12 conditions.

During the baseline recording and SD days, red light at very low intensity was present to allow the experimenters to observe the mice. Mice were sleep deprived for 2 hr according to the 'gentle handling' method.

## Mice for bioluminescence data collection

SKH1 mice (n = 6) were implanted with an osmotic mini-pump (Alzet, 1002, luciferin concentration: 35 mg/mL; blue flow moderator) 2 days before the habituation. At the end of the light phase (~ZT10-ZT12), mice were moved from their cage to the RT-Biolumicorder for 2–3 days of habituation in LD. They were housed for 2.5 days in DD, after which the 2hOnOff protocol was initiated at light onset (ZT0) under the preceding LD conditions. At the start of each SD, mice were moved from the RT-Biolumicorder and placed into a novel cage that was in the same room as the EEG-implanted mice. Fifteen minutes before the end of each SD, mice were brought back to their RT-Biolumicorder cage.

## Bioluminescence recordings in *Pkg1-Luc* mice

The adenoviral vector used in the experiment was constructed by cloning the cassette flanked by PacI sites from vector prLV1 (*Du et al., 2014*) into vector pCV100, an E1/E3-deleted replication-incompetent first-generation adenovirus vector based on human adenovirus serotype 5 (a variant of pGS66 [*Schiedner et al., 2000*] with an additional deletion of Ad5 nt 28133–30818). In this construct, firefly luciferase is expressed from a bidirectionally active, minimal *Pgk1* promoter (note: the other side of the promoter carries Renilla luciferase cDNA, which was not used/measured in the framework of this study). See *Supplementary file 1* for the construct sequence. Liver cells were transduced with the pCV100 vector (2.1e10 to 4.5e11 adenoviral particles) via tail vein injection using an illuminated restrainer according to *Saini et al., 2013* and *Sinturel et al., 2021*. Male C57BL/6J mice kept under LD 12:12 were injected 2–5 days prior to pump implantation, and pump was implanted 3 days prior to commencing experiment in DD in the RT-Biolumicorder. To allow passage of photons emitted from liver, mid-section of mice' backs was shaved.

## Data collection of sleep-wake state

### Simultaneous recording of sleep-wake state and PER2 bioluminescence

Batteries (hearing aid; Ansmann, 312 PR41, 1.45 V 180 mAh) were inserted into the NeuroLogger. This insertion was timed with the clock of the computer that controlled the RT-Biolumicorder to *post hoc* align the EEG/EMG signals with the bioluminescence signal. In addition, time stamps provided by the SyncBox (NeuroLogger, TSE) were used to verify the start and end time of the EEG/EMG recording. The cerebellar electrode was used as a reference for both EMG and EEG. Data were sampled at 256 Hz. The frontal signal was subtracted from the parietal signal (EDF Browser) to support sleep-wake state determination by enhancing the identification of slow waves and theta waves within the same trace. The data were subsequently loaded in Somnologica (Somnologica 3, MedCare) to determine offline the mouse's behavior as wakefulness, REM sleep, or NREM sleep per 4 s epochs based on the EEG and EMG signals. Wakefulness was characterized by EEG activity of mixed frequency and low amplitude, and present but variable muscle tone. NREM sleep (NREM) was defined by synchronous activity in the delta frequency (1–4 Hz) and low and stable muscle tone. REM sleep (REM) was characterized by regular theta oscillations (6–9 Hz) and EMG muscle atonia.

### 2hOnOff experiment

EEG and EMG signals were recorded continuously for 96 hr. The recording started at the beginning of the subjective rest phase, ZT0 of the preceding LD cycle. The analog EEG and EMG signals were amplified (2000×) and digitized at 2 kHz and subsequently down-sampled to 200 Hz and stored. Like the EEG and EMG traces obtained with the NeuroLogger, the data were imported in Somnologica and sleep-wake state was determined per 4 s epochs. LMA was monitored with passive infrared activity (Actimetrics) and recorded with ClockLab (Actimetrics).

### Data analysis

### Route of luciferin administration

Circadian time was determined to inspect the circadian changes in bioluminescence relative to LMA. To this end, the period length per mouse was determined based on activity measurements (1 min resolution) by chi-square analysis in ClockLab. Subsequently, the activity and bioluminescence data were folded according to the period. The activity data were binned per 10 min, and activity onset was visually determined for each mouse and set at CT12. The aligned activity and bioluminescence data were subsequently averaged per circadian hour across mice.

### Spontaneous sleep-wake state and bioluminescence

Bioluminescence and activity were sampled at a resolution of 4 s, which is the same resolution of the epochs for sleep-wake state determination. Data processing was subsequently performed in MATLAB 2017b (The MathWorks, Inc, Natick, MA) and R (version 4.0.0). Linear trends were removed from the signal by the build-in function *detrend* in MATLAB and R (*pracma* package). Subsequently, the bioluminescence signal was expressed relative to the overall mean per mouse to account for inter-individual differences. Changes in PER2 would be expected to occur at a slower rate than 4 s. Therefore, sleep-wake and bioluminescence data are averaged per blocks of 3 min.

For the sleep-wake transition analysis, 3 min intervals in which the mouse was awake for more than 50% (i.e., 23 or more 4 s epochs) were deemed awake, otherwise as asleep. Based on this new 3 min sleep-wake sequence, clear sleep-to-wake and wake-to-sleep transitions were selected according to the following criteria: at least 9 min (three 3 min intervals) of the initial state had to be followed by at least 15 min (five 3 min intervals) of the other state. Transitions were followed both forward and backward in time as long as state did not change. Bioluminescence for all 3 min intervals of a transition was expressed as a percentage of the level reached at the transition; that is, the average between the level reached in the last 3 min prior to the transition and the first 3 min after the transition. Transitions were aligned according to time of the state transition (time 0) and then averaged first within and then across mice. Average time courses were reported for the longest time spent in state after the transition to which all mice contributed.

### 2hOnOff experiment

Bioluminescence data obtained 5 min before and 10 min after the SDs were excluded from analysis. Subsequent data normalization of the bioluminescence data was as above. To determine the influence of the 2hOnOff protocol on the strength of the ongoing circadian oscillation of the circadian distribution of sleep and wake, as well as on PER2 bioluminescence, an estimation of amplitude by sinewave fitting was done (MATLAB, $fit = Y_0 + a * sin\left(\frac{2*pi}{b} * t + c\right)$).

## Modeling PER2 bioluminescence with a damped harmonic oscillator

The temporal dynamic of PER2 bioluminescence was modeled according to the equation of motion describing a driven damped harmonic oscillator:

$$\frac{d^2x}{dt^2} + \gamma\frac{dx}{dt} + \omega_0^2 x = F$$

where x is the displacement of the oscillator, γ is the damping constant of the model, and $\omega_0^2$ is the string constant defining the natural frequency of the model. The driving forces used in this model are the LMA representing waking $\overrightarrow{F}_{WAKE}$, and a circadian force $\overrightarrow{F}_{PERI}$, represented as a sinewave. The momentary force exerted on the oscillator is represented as the sum of these two forces (see *Figure 4A*):

$$F\left(t\right) = \beta LMA_t + A sin\left(\omega t + \varphi\right)$$

where $\beta$, $A$, and $\varphi$ are respectively the coefficient applied on LMA amount, the amplitude of the circadian sinewave force, and the phase of the circadian sinewave force. These coefficients were the free parameters to be optimized in the model. $\omega$ is the angular velocity of the sinewave and was set to 2*pi/23.7, based on the residual PER2-bioluminescence rhythm present in the baseline of SCNx mice. To solve this equation and optimize for parameters that best describe the observed PER2

bioluminescence, we proceeded as follows: the relative bioluminescence data from both experiments were averaged across mice using 30 min bins. LMA was averaged across mice using 6 min bins. LMA could not be measured directly during SDs and was estimated by assessing the increase in LMA during SDs measured in EEG-implanted mice, which was found to be 2.4 times higher compared to average baseline levels. Thus, the SD effect was estimated using 2.4 times the mean activity observed during baseline (i.e., 181.9 and 180.2 for the SCNx and 2hOnOff, respectively).

To solve the second-order ordinary differential equation (ODE) of the driven harmonic oscillator, we transformed it into the following system of two first-order ODEs describing the change of position and speed of our oscillator:

$$x_1' = x_2$$
$$x_2' = F - \gamma x_2 - \omega_0^2 x_1$$

We then used the fourth-order Runge–Kutta (RK4) numerical method to approximate the solution using a fixed time step of 0.1 hr. In the SCNx experiment, initial values of speed ($x_2\left(0\right)$) and position ($x_1\left(0\right)$) were set to 0. For the 2hOnOff experiment, the model was generated for 20 days prior experiment to reach a steady state using replication of LMA observed in baseline (T0–T24). The position and speed of the oscillator at the end of the 20 days were taken as initial values for the fitting. We optimized the model for the following parameters: intercept (equilibrium position of the oscillator), natural frequency, damping constant, and coefficient for the force exerted by LMA, amplitude of circadian force and phase of circadian force. We optimized the fitting by minimizing the RSS between predicted position of our model and the observed PER2-bioluminescence level. The box-constrained PORT routines method (nlminb) implemented in the optimx/R package (**Nash and Varadhan, 2011**) was used to minimize the RSS.

The goodness of fit of the model was assessed as follows. We assumed that the model errors follow a normal distribution and computed a BIC value for the model according to

$$BIC = n \ln\left(\frac{RSS}{n}\right) + k \ln\left(n\right)$$

where n is the number of observations, and k is the number of optimized parameters +1 of the model. We approximated the Bayes factor (BF) between our model and a flat model (linear model with intercept only) as follows:

$$BF \approx exp\left(-\frac{1}{2}\left(BIC_{flat} - BIC_{model}\right)\right)$$

To compute confidence interval of our model parameters, we used 500 Monte Carlo simulations and calculated a confidence interval for our parameters based on 95% empirical quantiles (95% CI). The method was adapted from the code of Marc Lavielle (Inria Saclay [Xpop] and Ecole Polytechnique [CMAP], Université Paris-Saclay, France) for nonlinear models, available here: http://sia.webpopix.org/nonlinearRegression.html.

## Statistics

Statistics were performed in R (version 4.0.0), SAS (version 9.4), and Prism (version 7.0), with the threshold of significance set at $\alpha$ = 0.05. Performed statistical tests are mentioned in the text and figure legends.

## Acknowledgements

We are greatly indebted to all who helped with the sleep deprivations: Lisa Härri, Charlotte Hor, and Jeffrey Hubbard, and especially to those who sacrificed their sleep during the graveyard shifts: Kostas Kompotis, Simone Mumbauer, Violeta Castelo-Szekely, and Sonia Jimenez. We also thank Sonia for her help with the sleep-wake annotation of EEG/EMG files. We thank David Gatfield for a great suggestion and David Gatfield and Florian Kreppel for designing, cloning, growing, and producing the adenoviral vector used for *Figure 1—figure supplement 5*. This study was performed at the University of Lausanne, Switzerland, and supported by the Swiss National Science Foundation (SNF

no. 146694 to PF supporting MMBH and SNF no. 179190 to David Gatfield supporting GK) and the State of Vaud (supporting MMBH, MJ, YE, and PF).

## Additional information

### Funding

| Funder | Grant reference number | Author |
| --- | --- | --- |
| Schweizerischer Nationalfonds zur Förderung der Wissenschaftlichen Forschung | 146694 | Marieke MB Hoekstra |
| Schweizerischer Nationalfonds zur Förderung der Wissenschaftlichen Forschung | 179190 | Georgia Katsioudi |
| State of Vaud | | Marieke MB Hoekstra Maxime Jan Yann Emmenegger Paul Franken |

The funders had no role in study design, data collection and interpretation, or the decision to submit the work for publication.

### Author contributions

Marieke MB Hoekstra, Conceptualization, Data curation, Formal analysis, Investigation, Project administration, Visualization, Writing – original draft, Writing – review and editing; Maxime Jan, Conceptualization, Data curation, Formal analysis, Investigation, Methodology, Visualization, Writing – review and editing; Georgia Katsioudi, Investigation, Methodology, Writing – review and editing; Yann Emmenegger, Investigation, Methodology; Paul Franken, Conceptualization, Data curation, Formal analysis, Funding acquisition, Project administration, Supervision, Visualization, Writing – review and editing

### Author ORCIDs

Marieke MB Hoekstra http://orcid.org/0000-0003-0723-2026
Maxime Jan http://orcid.org/0000-0001-6483-7430
Paul Franken http://orcid.org/0000-0002-2500-2921

### Ethics

All experiments were approved by the Ethical Committee of the State of Vaud Veterinary Office Switzerland under license VD 2743, 3201 and 3402.

### Decision letter and Author response

Decision letter https://doi.org/10.7554/eLife.69773.sa1
Author response https://doi.org/10.7554/eLife.69773.sa2

## Additional files

### Supplementary files

• Supplementary file 1. Sequence of the pCV100 viral vector construct containing *Pkg1-Luc*.

• Transparent reporting form

### Data availability

Data underlying the experimental figures is available through the source files.

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
