## [Editor Report]

This work contributes interesting data to both the circadian and sleep fields as it presents evidence that clock gene expression in peripheral tissues can be regulated in sleep-wake-state- and peripheral-circadian-dependent manners. To support this idea, the authors monitor sleep-wake state, as well as PER2 expression (utilizing a PER2-luciferase system), in both intact or SCN-lesioned freely behaving mice. Analysis of central and peripheral PER2LUC levels, under diverse sleep protocols, and aided by mathematical models allows them to support the idea that in peripheral tissues the clock gene circuitry integrates sleep-wake information, potentially contributing to behavioral adaptability to homeostatically respond to different challenges.

---

## [Decision Letter]

**Decision letter after peer review:**

[Editors’ note: the authors submitted for reconsideration following the decision after peer review. What follows is the decision letter after the first round of review.]

Thank you for submitting your work entitled "The sleep-wake distribution contributes to the peripheral rhythms in PERIOD-2" for consideration by *eLife*. Your article has been reviewed by 3 peer reviewers, and the evaluation has been overseen by a Reviewing Editor and a Senior Editor. The following individual involved in review of your submission has agreed to reveal their identity: Hiroki Ueda (Reviewer #1).

Our decision has been reached after consultation between the reviewers. Based on these discussions and the individual reviews below, we regret to inform you that your work will not be considered further for publication in *eLife*.

As you can see from the feedback provided by the reviewers they found the data interesting and thought-provoking, yet they also raised substantial concerns about different aspects of the work. Importantly, the reviewers noted that some of the main conclusions (based on the presented data) are not fully supported. A considerable amount of work that is beyond the scope of a revision at *eLife* would be needed to address these concerns.

*Reviewer #1:*

In this paper entitled "The sleep-wake distribution contributes to the peripheral rhythms in PERIOD-2", the authors observed PER2 expression in peripheral tissues of freely behaving animals and found that the sleep-wake distribution contributes to the peripheral PER2 dynamics. In addition, implementing the mathematical model, they showed that the peripheral PER2 rhythms can be generated by two factors: a sleep-wake state dependent one and a peripheral (not master) circadian dependent one. Despite the difficulty to dissect the effect of these two factors in vivo, the authors, dealing with this problem by the unique experimental designs, provided direct evidence that sleep-wake state itself affects peripheral daily rhythms. However, this paper has a couple of points to be improved, which are summarized in the comments below.

1. As for Figure 1B, 1C and 1D, the authors showed PER2 bioluminescence changes according to circadian rhythm and state transitions in only one individual mouse (ID: 2MP82), which might be insufficient for concluding that the PER2 bioluminescence changes depending on the circadian and sleep-wake cycle. Plus, there might be a mistake in the time label of Figure 1B (is this plot started from time = 6 or time = 0?). Thus, the reviewer suggests that the authors should add data about other mice and perform the statistical analyses to clarify the validity of the circadian- and sleep-wake-dependent change of PER2 bioluminescence. With Figure 4, the reviewer suggests the same, i.e., the authors should analyze not only one representative mouse but a group of mice.

2. In lines 202 to 205, the authors concluded that the behavioral rhythm reduction by 2hOnOff protocol appeared to be insufficient to consistently reduce the amplitude of the PER2-bioluminescence oscillation. However, it is difficult to follow its logic toward the conclusion. Thus, the reviewer recommends that the authors should describe their logic in more detail.

3. As for Figure 4 and Table 2, the authors compared two mathematical models and concluded that the two forces model has better prediction accuracy. However, it is unclear whether the circadian force itself or the combination of the two forces is important to predict PER2-bioluminescence dynamics. The reviewer suggests that the authors should add a comparison with a model that using a single circadian force (F = FPERI).

4. In lines 277 to 280, the authors described the increase in PER2 amplitude during the 2hOnOff experiment by advancing the phase of FPERI by 6h. The reviewer suggests that the authors should clearly explain why they apply their mathematical model to the 6h advancing, although they observed 1-2h phase advance in their experiments.

*Reviewer #2:*

By measuring EEG/EMG and PER2::LUC bioluminescence (most signal comes from the kidney) from freely behaving mice simultaneously or from separate cohort mice with similar conditions, the authors observed that sleep state acutely changed PER2::LUC bioluminescence, repeated daily sleep deprivation reinstated circadian PER2::LUC rhythm in SCN ablated behaviorally arrhythmic mice. The authors also did a mathematical simulation modeling peripheral PER2::LUC bioluminescence influenced by a sleep dependent force and a SCN independent force.

I believe the data presented in the current manuscript, however, I don't think the main conclusions are supported by the data. The acute changes (within 30 min) of luminescence they observed is most likely caused by in vivo factors (Not changed in PER2::LUC protein, artifact). It is very problematic to assess the amplitude change from averaged luminescence record from multiple behaviorally arrhythmic mice (analysis flaw). Although the above mentioned two major conclusions must be removed from the manuscript, the study validates previously published results done by ex vivo tissue and in vivo anesthetized animal with luminescence measure from freely behaving mice. So I encourage the authors to publish the data to another journal (please also see my specific points below), but I don't think discovery reported in the work is novel for the publication in *eLife*.

1. It is unlikely that the acute change of PER2::LUC luminescence reflects PER2 protein translation and degradation. As the authors refer to correctly, luminescence from PER2::LUC knockin mice reflects PER2::LUC fusion protein (not Per2 transcription). The authors didn't correctly refer to the studies measuring light induction of Per2 mRNA and protein in the SCN. The light induction of Per2 mRNA in the SCN can only be seen 60 min after onset of the light pulse (no induction was observed at 30 min after the light pulse. Takumi et al. 1998 and many other publications). Increased PER2 protein in the SCN can only be seen at 4 h after the light pulse (no change was observed 2 h after the light pulse. Yan and Sliver 2004, Antoun et al., 2012). Therefore, the acute luminescence change observed at sleep wake transitions (Figure 1D) is unlikely reflecting changes in the PER2::LUC fusion protein. Temperature, pH, and Ca/Mg and ATP levels all strongly affect luminescence produced by firefly luciferase and luciferin. It was shown that the ultradian rhythm of the core body temperature closely matched the ultradian behavior rhythm (Blessing 2018 PMID: 30454601). The authors referred to their own published study that luminescence rhythm change was vilified by qPCR. However, the time scale of their published work showed that changes occurred after 6 h of sleep deprivation, not in the 30 min range.

2. The advantage of continuous luminescence measurement from the same mouse is that the presence and absence of the rhythm can be determined from a mouse that has no phase marker (e.g. SCN lesioned mice). Arrhythmicity (or lowered amplitude of the rhythm) observed by conventional population sampling (from multiple mice) cannot be determined if individual mice lose rhythmicity, or if individual mice retained intact rhythm but they are out of sync. From the data presented in Figure 2, the authors claimed repeated sleep deprivations reinstated the rhythm, but they didn't analyze amplitude change before, during, and after sleep deprivation in individual mice (even they didn't provide individual data anywhere else, they provided individual data in other experiments). It is well known that the SCN-lesioned mice do not show a daily rhythm under LD cycle (Light cannot reinstate behavior rhythm). The authors stated that they performed daily sleep deprivation to the mice 5 weeks after SCN-lesioned. From average luminescence rhythms of 4 SCN-X mice, the authors cannot conclude their main claim that sleep deprivation reinstated the peripheral circadian rhythm. It is possible there is no change in the amplitude of individual peripheral circadian oscillators, but repeated sleep deprivation entrained those rhythms (the authors averaged individual luminescence relative to the phase of daily sleep deprivation).

3. In the complementary experiment presented in Figure 3, the authors analyzed the amplitude changes from individual mice, but didn't analyze from the averaged data. The authors observed a reduced amplitude by 2hOnOff procedure in 4 out of 6 mice, and an increased amplitude in 2 out of 6 mice. But the authors concluded that the 2hOnOff procedure did not "consistently" affect the rhythm (in another place, they wrote "2hOnOff protocol didn't noticeably affect the ongoing circadian PER2 oscillation").

4. Because of the issues pointed out above, the parameters used for the mathematical simulation are not valid. Use of locomotor activity as a wake parameter is also not appropriate. Mice can be wake without moving.

5. In the method section, the authors described each procedure. However, it was impossible for me to know how exactly each individual mouse went through the procedures. It seems some sleep deprivation was done under dim red light and other was done under light. I suggest the authors provide a timeline (e.g. flowchart figure) of procedures in each experiment.

6. Since in vivo bioluminescence recording became popular technology, I strongly suggest the authors move the data presented in supplementary Figure 2 to the main text as an actual figure. Were CAG mice also crossed with hairless strain? Does tissue explanted from this mouse in vitro not show a circadian rhythm? It even looks like luminescence from the mice with luciferin supplemented by osmic pump shows a weak rhythm. I wonder if CAG promoter may have circadian rhythm as CMV promoter does (Collaco et al., 2005). Amplitude of activity rhythm is lowered in mice with pump. Is this due to surgery (pump implantation)? If so, lowered amplitude of luminescence rhythm might also be affected by surgery. Having a control (saline in pump and luciferin supplied through drinking water) is necessary.

*Reviewer #3:*

This is an interesting study. As far as I know, there are no previous studies performing simultaneous measurement of sleep-wake state and bioluminescence (clock gene expression) in freely moving mice. In addition, their measurement method was well validated through several pilot studies, and measurements obtained using this method are therefore reliable. In fact, the presented data are clear and convincing. However, although I have no doubt that the results are useful and valuable for a wide range of circadian scientists, I feel the present version of their work may not be a strong candidate for publication in this journal because of insufficient novelty and impact.

1) The results clearly revealed the temporal relationship between sleep-wake state and PER2 bioluminescence through simultaneous monitoring in freely moving mice, with high time resolution. However, because previous studies have already found that sleep deprivation results in a change in PER2 levels, I feel that the present study does not make obvious progress toward further understanding the mechanism. The simultaneous monitoring experiments are interesting but each of the techniques was originally established by other groups. For example, some findings related to the identification of systemic or molecular pathways controlling PER2 expression during sleep deprivation may be required.

2) Although a detailed description of the method for sleep deprivation is important, I was not able to find it in the main text. The authors should clarify whether sleep deprivation itself or other factors such as stress signals triggered by sleep deprivation affect PER2 expression. At this time, I feel not sleep deprivation per se but some stress signals affect PER2 expression. Many previous studies have shown that glucocorticoids, well-known stress hormones, strongly affect peripheral clocks. Sleep deprivation, one of the stimuli activating stress signals, may therefore affect PER2 expression in peripheral clocks via glucocorticoid signaling pathways. The contribution of glucocorticoids to PER2 expression during sleep deprivation can be examined simply through surgical removal of the adrenal glands.

3) Their conclusions may be specific to the kidney. All bioluminescence data were obtained through monitoring only the kidney. The authors should justify why they can conclude that sleep deprivation affects "peripheral clocks" only judging from kidney data. To generalize kidney data to other peripheral clocks, they should repeat similar experiments using at least another organ or tissue. Real-time monitoring of bioluminescence from the liver is technically simple because injected adenoviruses (carrying a reporter gene driven by a clock gene promoter) specifically infect the liver.

[Editors’ note: further revisions were suggested prior to acceptance, as described below.]

Thank you for resubmitting your work entitled "The sleep-wake distribution contributes to the peripheral rhythms in PERIOD-2" for further consideration by *eLife*. Your revised article has been evaluated by Kate Wassum (Senior Editor) and a Reviewing Editor.

The manuscript has been improved but there are some remaining issues that need to be addressed, as outlined below:

As you will be able to see in the accompanying comments, while reviewers valued the changes included in this new version of the MS, they still raised important questions. Some may be addressed by reanalyzinging the data already present in the paper (non-circadian reporter mice, CAG-luc). It is claimed that PER2LUC luminescence spikes in Figure 1B are caused by changes in sleep-wake status (analysis is shown in Figure 1C). Currently, for the CAG-LUC animals, there is only group average data (Supplementary Figure 3B). A valid suggestion, hinted by reviewers, is to show individual records: if spikes are not seen in these animals it helps excluding the possibility that the in vivo luminescence spikes are an artifact (e.g. caused by core body temperature changes). Although it doesn't seem to be recorded EEG/EMG from the non-circadian reporter mice, a similar analysis (based on defining sleep as sustained immobility) could be conducted.

Yet, what would really provide strong support to the claims, eliminating doubts about potential artifacts, is to validate the data with conventional methods (immunoblot). As it is mentioned in the MS, during previous work (Curie 2015, PMID: 25581923) the authors have partially validated by Western blot analyses peripheral Per2luc bioluminescence data (Figure S4, liver). Yet, the temporal resolution, and experimental condition of such analyses may not be fully comparable to what is present here,

It is also important to consider that most of the conclusions drawn from the data are based on the temporal correlation between kidney-derived bioluminescence and sleep-wake states. Therefore, it is difficult to generalize the temporal correlation between single organ-derived bioluminescence and sleep-wake states to other peripheral and central clocks. Such limitation of the work (in the absence of additional data) should be clearly acknowledged.

*Reviewer #1:*

In this paper entitled "The sleep-wake distribution contributes to the peripheral rhythms in PERIOD-2", the authors attempted to find the factor(s) directly contributing to clock gene expression in peripheral tissues. To achieve that, they observed PERIOD-2 (PER2) expression in peripheral tissues of freely moving mice and developed new mathematical models to analyze them. They found that the sleep-wake distribution correlates to the peripheral PER2 dynamics, and that the peripheral PER2 rhythms can be predicted by sleep-wake-state- and peripheral-circadian-dependent factors. Despite the difficulty to dissect the effects of circadian and the sleep-wake cycle on gene expression in vivo, they provided direct evidence that the sleep-wake cycle itself affects peripheral daily rhythms by the unique experimental designs. However, there are still a few points to be addressed before publication including a detailed validation of the PER2 bioluminescence system, which would further support their conclusion.

All things considered, their findings will have an impact on the chronobiology field, as giving direct evidence on sleep-wake-state-dependent regulation of peripheral clock gene expression. In addition, those methods can be useful to investigate the gene expression not only in the kidney or cerebral cortex but also in other various tissues of freely moving mice.

In the revised paper, the authors addressed almost all our concerns with proper correction and additional analyses, but it has some points to be improved before publication, which are summarized in the comments below.

1. The authors assumed that PER2 bioluminescence directly reflected expression of PER2 protein, but it has not been fully confirmed yet especially in the fast (within an hour) dynamics. For further progress of the method, the combination with other techniques to quantify fast and small (< 10%) protein changes would be required, as the authors mentioned in the paper.

2. This reviewer suggests that the authors should discuss about the controversial results regarding PER2-bioluminescene amplitude and sleep-wake distribution in Figure 3 and Suppl. Figure 7. When the amplitude of the sleep-wake distribution was reduced, PER2-bioluminescence amplitude was reduced in 4/6 mice as the model expected, while it was increased in 2/6 mice.

3. Regarding Figure 3E, the authors showed that the 2hOnOff protocol decreased the amplitude of sleep-wake distribution. However, the sleep-wake distribution during the recovery phase was not assessed. Thus, the reviewer recommends that the authors should add data of recovery day to Figure 3E and compare it to the PER2 bioluminescence trends (Figure 3C).

4. In lines 290 to 294, the authors concluded that the prediction from the model, the behavioral rhythm reduction reduces PER2 bioluminescence amplitude, which is consistent with their observation in the Figure 3. However, this conclusion could be difficult to understand because the 2hOnOff protocol increases the PER2 bioluminescence amplitude in one-third of mice (there may be no significant change between BSL and the 2hOnOff protocol; see Figure 3C). The authors provided an additional model (Suppl. Figure 7A-C) to explain the individual difference, and thus the reviewer suggests the authors to describe that conclusion after the modeling part to make the logic clearer.

5. Regarding Suppl. Figure 7A-C, the authors described the increase in PER2 amplitude during the 2hOnOff experiment by advancing the phase of FPERI more than 2.5h. The reviewer suggests that the authors could mention these results in Results or Discussion section.

*Reviewer #2:*

Hoekstra et al. investigated the relationship between sleep state and PER2::LUC luminescence in freely behaving mice and showed that luminescence acutely changed at sleep/wake transitions. The authors also showed that daily sleep deprivation enhanced the PER2::LUC luminescence rhythm amplitude in SCN ablated mice.

A strength of this study is the simultaneous recording of EEG/EMG/activity and PER2::LUC luminescence in freely moving mice.

Weakness: One obvious concern is that acute PER2::LUC luminescence changes occurring at sleep/wake transitions are artifacts (e.g. due to acute change in core body temperature), but not caused by synthesis/degradation of PER2 protein. Because those luminescence changes are relatively large and fast (30-40% change within 30 min), a conventional method may be used for validation. The authors recorded activity and bioluminescence from non-circadian luminescence reporter mice (CAG -luc). Alternatively, the authors can analyze that data and exclude artifact possibility if luminescence change doesn't occur at active/inactive transitions.

Another weakness is the PER2::LUC luminescence measurement from the brain. The authors must show IVIS imaging and show the signal coming from the brain and not from other tissues.

Abstract: Line 18, 19: instead of saying PER2 increases or PER2 rhythmicity, I suggest using PER2::LUC luminescence increases, PER2::LUC luminescence rhythmicity.

Line 89, Suppl. Figure 1: should be Suppl. Figure 3.

Figure 1B: The hypnogram only includes sleeping and waking, please include REM and NREM. I understand that the authors only wanted to indicate sleep/wake transition used for the luminescence analysis indicated in C, but I think the authors should plot standard hypnogram.

Figure 1C: To avoid confusion, it is better to use the same colors as in A and B (pink: peripheral, purple: central).

Figure 2: The authors should show individual records in the supplemental information. I know the authors discussed two possibilities that can explain changes in amplitude in group average data. The amplitude of an individual mouse was shown in Figure 2C. But I think, like me, the readers will still want to see the individual record. The amplitude of 3 cycles in REC should be analyzed separately (Plot as REC day 1, REC day 2 and REC day 3). Technically REC day 1 is the same as SD, but it is good to show how the amplitude in the group average and the individual mouse declines.

Line 573-584: This is an important discussion addressing the possible temperature artifact. However, all statements are based on analysis, "data not shown", unfortunately. The authors must show the data.

Line 763: I think the activity data was binned, not averaged.

Supplementary Figure 1: I know luminescence recording was done in DD, but please indicate this, as the lighting condition in the other treatments was stated.

Supplementary Figure 3B: Please indicate individual records as well, so the relationship between luminescence and activity can be evaluated.

*Reviewer #3:*

Response to the comment #1

It is my understanding that almost all of the conclusions made by the authors are based on the temporal correlation between kidney-derived bioluminescence and sleep-wake states. Therefore, I do not feel convinced about these conclusions without additional experimental evidence suggesting the causal link between spontaneous wakefulness and PER2 expression.

A few previous studies, although not cited in the present version, clearly demonstrated that arousal levels affect the circadian clock via the SCN (for example, PNAS 2016 Yamakawa el al). I think that the present data may not make obvious progress from these previous achievements.

Response to the comment #3

The authors claim that spontaneous wakefulness-driven PER2 increase is one of the major findings. Therefore, under not sleep deprivation but spontaneous wakefulness conditions, they need to confirm PER2 expression in other organs and apply expression data to their mathematical model.

Many previous studies have indicated that circadian phase in clock gene expression varies among peripheral organs and tissues, and it is therefore difficult to generalize the temporal correlation between single organ-derived bioluminescence and sleep-wake states to other peripheral and central clocks. In particular, circadian phase of the SCN is largely phase-advanced in comparison with that of peripheral clocks, raising a question on the author's thought that spontaneous wakefulness is a similar trigger of PER2 in both peripheral and central clocks.

[Editors’ note: further revisions were suggested prior to acceptance, as described below.]

Thank you for resubmitting your work entitled "The sleep-wake distribution contributes to the peripheral rhythms in PERIOD-2" for further consideration by *eLife*. Your revised article has been evaluated by Kate Wassum (Senior Editor) and a Reviewing Editor.

The manuscript has been improved but there are some remaining issues that need to be addressed, as outlined below:

There are only two final pending issues that we anticipate should be easy to incorporate as part of the results/discussion and be included in the final rebuttal letter, without the need for any additional experiments.

Essential revisions:

1) Although the new data has clearly enriched the paper, this still raises the concern of one of the reviewers regarding the presence of low amplitude baseline rhythm in luminescence (particularly as CAG-luc appears to be activated by wake: the reviewer expects to see a basal circadian change of luminescence in CAG-luc mice).

2) Without showing an absolute luminescence value emitted from the brain, liver, and kidney in the same mouse, it cannot be utterly said that the luminescence signal from the entire mouse is just coming from the cortex. Part of this concern is based on the fact that the Curie et al., 2015 work not only shows signals (in B6 mice) coming from the cortex, but also clear signals coming from the kidney/liver (as visualized with the IVIS signals, whereas the RT-Biolumicorder does not allow sorting those signals).

*Reviewer #2:*

Hoekstra et al. investigated the relationship between sleep state and PER2::LUC luminescence in freely behaving mice and showed that luminescence acutely changed at sleep/wake transitions. The authors also showed that repeated daily sleep deprivation reinitiated the PER2::LUC luminescence rhythm in SCN ablated behaviorally arrhythmic mice. This re-initiated rhythm sustained a few cycles in constant conditions, indicating that sleep can synchronize (or coordinate) peripheral circadian oscillators.

A strength of this study is the simultaneous recording of EEG/EMG/activity and PER2::LUC luminescence in freely moving mice.

One obvious concern is that acute PER2::LUC luminescence changes occurring at sleep/wake transitions are artifacts (e.g. due to acute change in core body temperature, Calcium/Magnesium, pH, ATP), but not caused by the synthesis/degradation of PER2 protein. After two major revisions, the authors now provided the data showing that no luminescence change occurs at sleep/wake transitions in Pkg1-luc activity (housekeeping gene promoter activity measured by luminescence reporter). Although it is still difficult for this reviewer to believe 30-40% changes in PER2 protein can occur within 30 min in the brain and peripheral organs, the new Pkg1-luc data rejects the above-mentioned concern. The authors also provided an interesting new analysis that acute luminescence changes at sleep/wake transitions in CAG-luc mice are much faster compared to changes in PER2::LUC mice. The authors reasoned this by saying the CAG synthetic promoter is activated by transcription factors (e.g. NF-κB, CREB, AP-1). This is an interesting result but raises a new concern. The authors used this CAG-luc mouse to optimize the luciferin administration method, as this is a non-circadian reporter. However, if being awake activates CAG promoter, this reviewer expects to see a basal circadian change of luminescence in CAG-luc mice.

Another relatively minor concern is that the authors used PER2::LUC in pigmented C57BL/6 mice with a cranial window for "the central quantification of bioluminescence". This reviewer suggested the authors to perform IVIS imaging and show the signal coming from the brain and not from other tissues. Instead, the authors referred to their previous publication (Curie et al., 2015) that the luminescence signal is from the cortex. However, in this paper, the authors actually detected a signal from the liver and kidney as well. Because the authors only showed the hold changes in those tissues (brain, liver kidney), it is not known how strong the signal from cranial windows compare to that from the liver and kidney. Without showing an absolute luminescence value emitted from the brain, liver, and kidney in the same mouse, the authors cannot say that the luminescence signal from the entire mouse is from the cortex.

This reviewer appreciates the efforts of the authors to address the concerns raised by the reviewer.

This reviewer can now recommend this work to be published in this journal. However, those two concerns need to be addressed.

One additional suggestion is that it may not be clear to the reader that mice with a cranial window is used for measuring signal from the cortex. This is only mentioned in the methods section. It may be better to state this in the main text and figure caption.

---

## [Author Response]

[Editors’ note: the authors resubmitted a revised version of the paper for consideration. What follows is the authors’ response to the first round of review.]

Reviewer #1:In this paper entitled "The sleep-wake distribution contributes to the peripheral rhythms in PERIOD-2", the authors observed PER2 expression in peripheral tissues of freely behaving animals and found that the sleep-wake distribution contributes to the peripheral PER2 dynamics. In addition, implementing the mathematical model, they showed that the peripheral PER2 rhythms can be generated by two factors: a sleep-wake state dependent one and a peripheral (not master) circadian dependent one. Despite the difficulty to dissect the effect of these two factors in vivo, the authors, dealing with this problem by the unique experimental designs, provided direct evidence that sleep-wake state itself affects peripheral daily rhythms. However, this paper has a couple of points to be improved, which are summarized in the comments below.

We thank the reviewer for his positive evaluation and particularly for stating that the results “provided direct evidence that sleep-wake state itself affects peripheral daily rhythms”, which is the central claim of our work. We also thank the reviewer for the suggestions to help improve our manuscript.

1. As for Figure 1B, 1C and 1D, the authors showed PER2 bioluminescence changes according to circadian rhythm and state transitions in only one individual mouse (ID: 2MP82), which might be insufficient for concluding that the PER2 bioluminescence changes depending on the circadian and sleep-wake cycle. Thus, the reviewer suggests that the authors should add data about other mice and perform the statistical analyses to clarify the validity of the circadian- and sleep-wake-dependent change of PER2 bioluminescence. With Figure 4, the reviewer suggests the same, i.e., the authors should analyze not only one representative mouse but a group of mice.

Indeed, Figure 1B and -C (now Figure 1B) represented an example of one mouse. Group means based on 5 mice were, however, depicted in Figure 1A and -D (now Figure 1A and C), as was mentioned in the figure legend, illustrated by the individual recordings of the 4 remaining mice in Suppl. Figure 3 (now Suppl. Figure 4), stated in the ‘transparent reporting’ form in our submission, and can be deduced from the raw data we provided for this and all other figures. The same holds true for Figure 4 where the model was based on recordings in 4 and 6 mice, respectively. In response to a suggestion from Reviewer 2, we have included a supplementary figure (Suppl. Figure 1) with an overview of all experiments explicitly stating the number of animals used in each.

Plus, there might be a mistake in the time label of Figure 1B (is this plot started from time = 6 or time = 0?).

The plot starts at 12h, at a time of day that under the previous LD cycles is referred to as ZT12; i.e., the beginning of the dark phase. The tick and label indeed disappeared. This has been corrected in the revised version.

2. In lines 202 to 205, the authors concluded that the behavioral rhythm reduction by 2hOnOff protocol appeared to be insufficient to consistently reduce the amplitude of the PER2-bioluminescence oscillation. However, it is difficult to follow its logic toward the conclusion. Thus, the reviewer recommends that the authors should describe their logic in more detail.

We thank the reviewer for this suggestion. With the use of the word ‘appeared’ we hoped it was clear that this was an interpretation. Our logic was as follows: If the sleep-wake distribution contributes to the diurnal changes in PER2 bioluminescence then we can expect that after experimentally reducing the amplitude of the sleep-wake distribution, the amplitude of the PER2 oscillation will be reduced as a result. With the 2hOnOff protocol we reduced, but did not eliminate, the amplitude of the sleep-wake distribution. This reduction was however accompanied by a reduction of PER2 amplitude in only 4 of the 6 mice and we therefore concluded that the reduction in sleep-wake amplitude appeared to be insufficient to elicit the anticipated PER2 effects in all mice. We now have eliminated this interpretative sentence from the Result section.

3. As for Figure 4 and Table 2, the authors compared two mathematical models and concluded that the two forces model has better prediction accuracy. However, it is unclear whether the circadian force itself or the combination of the two forces is important to predict PER2-bioluminescence dynamics. The reviewer suggests that the authors should add a comparison with a model that using a single circadian force (F = FPERI).

Again, we thank the reviewer for this valuable suggestion. We have now performed the suggested additional analysis in the revised manuscript, illustrating better that both forces are needed and their combination represents an important statistical gain in predicting PER2. The results have been added to Figure 4B and C and are presented in the text (l. 326-332). We now extended the modeling to predict results of the 6h SD experiments using the parameters obtained in the other two experiments (Figure 4D, Suppl. Figure 7D, Table 2, l. 297-302).

4. In lines 277 to 280, the authors described the increase in PER2 amplitude during the 2hOnOff experiment by advancing the phase of FPERI by 6h. The reviewer suggests that the authors should clearly explain why they apply their mathematical model to the 6h advancing, although they observed 1-2h phase advance in their experiments.

This analysis was meant as thought-experiment to illustrate on the one hand the power of our dynamic model to perform *in silico* experiments and, on the other hand, that the initial conditions could give rise to diametrically opposing results. We made this clearer in the manuscript. We have now regrouped this analysis together with the analysis mentioned under the preceding point 3, as one of three questions addressed by the model as *in silico* experiments (l. 356-379). In a re-analysis for the revised manuscript, we found that the same effect (i.e., an increase of PER2 bioluminescence amplitude instead of the expected decrease) could already be observed with a 2.5h phase advance. We have added this finding to the revised manuscript (l. 360-365) and illustrated this analysis in Suppl. Figure 7C.

Reviewer #2:By measuring EEG/EMG and PER2::LUC bioluminescence (most signal comes from the kidney) from freely behaving mice simultaneously or from separate cohort mice with similar conditions, the authors observed that sleep state acutely changed PER2::LUC bioluminescence, repeated daily sleep deprivation reinstated circadian PER2::LUC rhythm in SCN ablated behaviorally arrhythmic mice. The authors also did a mathematical simulation modeling peripheral PER2::LUC bioluminescence influenced by a sleep dependent force and a SCN independent force.I believe the data presented in the current manuscript, however, I don't think the main conclusions are supported by the data. The acute changes (within 30 min) of luminescence they observed is most likely caused by in vivo factors (Not changed in PER2::LUC protein, artifact). It is very problematic to assess the amplitude change from averaged luminescence record from multiple behaviorally arrhythmic mice (analysis flaw). Although the above mentioned two major conclusions must be removed from the manuscript, the study validates previously published results done by ex vivo tissue and in vivo anesthetized animal with luminescence measure from freely behaving mice. So I encourage the authors to publish the data to another journal (please also see my specific points below), but I don't think discovery reported in the work is novel for the publication in eLife.

We have addressed each of the issues mentioned in detail below. What the reviewer refers to as “very problematic” and “analysis flaw” could be easily addressed. Moreover, the reasons given to deem our bioluminescence signal as artifact are flawed.

1. It is unlikely that the acute change of PER2::LUC luminescence reflects PER2 protein translation and degradation. As the authors refer to correctly, luminescence from PER2::LUC knockin mice reflects PER2::LUC fusion protein (not Per2 transcription). The authors didn't correctly refer to the studies measuring light induction of Per2 mRNA and protein in the SCN. The light induction of Per2 mRNA in the SCN can only be seen 60 min after onset of the light pulse (no induction was observed at 30 min after the light pulse. Takumi et al. 1998 and many other publications). Increased PER2 protein in the SCN can only be seen at 4 h after the light pulse (no change was observed 2 h after the light pulse. Yan and Sliver 2004, Antoun et al., 2012). Therefore, the acute luminescence change observed at sleep wake transitions (Figure 1D) is unlikely reflecting changes in the PER2::LUC fusion protein. Temperature, pH, and Ca/Mg and ATP levels all strongly affect luminescence produced by firefly luciferase and luciferin. It was shown that the ultradian rhythm of the core body temperature closely matched the ultradian behavior rhythm (Blessing 2018 PMID: 30454601). The authors referred to their own published study that luminescence rhythm change was vilified by qPCR. However, the time scale of their published work showed that changes occurred after 6 h of sleep deprivation, not in the 30 min range.

First, we have to correct two of the reviewer’s statements: (1) We did not claim that the literature showed that 30min light-pulses induce PER2 in the SCN and (2) we previously used western blot, and not qPCR, to validate the sleep-deprivation induced increases in PER2 bioluminescence in brain and liver.

We are aware that no study has reported on such quick induction of PER2 after a light pulse. We did, however, not study light effects in the SCN, and the effects of wakefulness on PER2 in peripheral tissues are vastly under-explored and might differ from those of light in the SCN as a variety of molecular pathways activated by waking are known to affect PER2 (reviewed in the Discussion). The fact that in the literature there is no precedent of wakefulness rapidly inducing small but significant changes in PER2 using more conventional techniques, cannot be used as argument that we reported on a mere artefact. We already cited the study by Cao *et al.* (Nat Neurosci 2015), demonstrating that PER2 protein can indeed be rapidly induced through other means than light; i.e., within 30 min after a serum shock (see Discussion paragraph l. 400-414). The wake-related rapid induction of PER2 is an important novel aspect of our study. Whether the signal obtained with the PER2::Luciferase construct indeed reflects PER2 protein was already discussed in detail in the Discussion (“Do changes in bioluminescence reflect changes in PER2 levels?”).

Although the variables listed by the reviewer do affect the bioluminescence reaction, they seem not to play a large role (see Discussion section: ‘Do changes in bioluminescence reflect changes in PER2 levels?’). The reviewer’s example of a potential temperature effect can illustrate this. Body temperature changes in a circadian fashion, yet the bioluminescence time courses reporting on clock genes known to be regulated in anti-phase, remain anti-phasic and do not simply follow the temperature dynamics. Also the lack of a noticeable rhythm in the CAGbioluminescence signal (Suppl. Figure 3B) confirms that the circadian rhythm in activity and associated temperature changes have little or no effect (or even decreases when activity is highest, see our response to point 6 below). We have added this observation to the Discussion (l. 577-579). While temperature increases quickly (within minutes) after awakening and stays high while awake (see the Blessing *et al.* reference mentioned by the reviewer and our own work: Hoekstra *et al. eLife* 2019), PER2-bioluminescence follows a much slower time course and initially continues to decrease before subsequently increasing (see Figure 1C). For this rebuttal we followed both central (see also comment 3 of Reviewer 3) and peripheral PER2-bioluminescence for as long as waking prevailed as opposed to the 1st 30min after wake onset only. The new analysis shows that peripheral PER2 increases and then again decreases with increasing wake duration, while central levels monotonically keep increasing consistent with the western-blot verified tissue-specific effects of experimentally prolonging wakefulness (Figure 1C and brain vs. liver in Curie *et al.* 2015). These results further underscore that changes in PER2 bioluminescence do not follow wake related changes in temperature. We have replaced the transition analyses in Figure 1C with this new analysis.

2. The advantage of continuous luminescence measurement from the same mouse is that the presence and absence of the rhythm can be determined from a mouse that has no phase marker (e.g. SCN lesioned mice). Arrhythmicity (or lowered amplitude of the rhythm) observed by conventional population sampling (from multiple mice) cannot be determined if individual mice lose rhythmicity, or if individual mice retained intact rhythm but they are out of sync. From the data presented in Figure 2, the authors claimed repeated sleep deprivations reinstated the rhythm, but they didn't analyze amplitude change before, during, and after sleep deprivation in individual mice (even they didn't provide individual data anywhere else, they provided individual data in other experiments).The authors stated that they performed daily sleep deprivation to the mice 5 weeks after SCN-lesioned. From average luminescence rhythms of 4 SCN-X mice, the authors cannot conclude their main claim that sleep deprivation reinstated the peripheral circadian rhythm. It is possible there is no change in the amplitude of individual peripheral circadian oscillators, but repeated sleep deprivation entrained those rhythms (the authors averaged individual luminescence relative to the phase of daily sleep deprivation).

We thank the reviewer for raising this issue. We had provided the individual amplitude information for the baseline bioluminescence rhythms of the SCNx mice in the first version of our manuscript (Suppl. Figure 5C and Figure 2B), but indeed did not include this information during and after the sleep deprivations. As mentioned in our reply to minor issue 1 from Reviewer 1, we have added these data to main Figure 2C. The results clarified that increased amplitude during the sleep deprivations does not simply result from a phase alignment as it is significantly larger than in the baseline recordings. This analysis and conclusion has now been added in the Results (l. 206-210).

It is well known that the SCN-lesioned mice do not show a daily rhythm under LD cycle (Light cannot reinstate behavior rhythm).

We did not make this claim in the manuscript and therefore fail to see the relevance of this comment. Moreover, the literature indicates that this claim is less well-known as the reviewer makes us believe (specifically addressed in Redlin and Mrosovsky J Comp Physiol A 1999, reviewed by Redlin Chronobiol Int 2001). Light is an important driver of rhythmic sleep-wake behavior (see e.g. Hubbard *et al.* PNAS 2021 *in press*; bioRxiv).

3. In the complementary experiment presented in Figure 3, the authors analyzed the amplitude changes from individual mice, but didn't analyze from the averaged data. The authors observed a reduced amplitude by 2hOnOff procedure in 4 out of 6 mice, and an increased amplitude in 2 out of 6 mice. But the authors concluded that the 2hOnOff procedure did not "consistently" affect the rhythm (in another place, they wrote "2hOnOff protocol didn't noticeably affect the ongoing circadian PER2 oscillation").

We did analyze the amplitude changes at the group level by comparing PER2 amplitude under baseline with the 2hOnOff regime, and mentioned the results in the figure legend (“resulting in overall lack of an effect of the 2hOnOff protocol”) along with the statistics. In the Results section, the lack of an overall effect was reiterated twice. We now have added the mean values and the SEMs to the figure legend for additional clarity.

4. Because of the issues pointed out above, the parameters used for the mathematical simulation are not valid.

The reviewer claims the parameters for the simulation are not valid, but does not explain how he or she arrives at this conclusion. Many issues were raised in the preceding three points, none of which make reference to the model. We will not try and guess what the reviewer might have meant especially since we have now addressed the issues pointed out above.

Use of locomotor activity as a wake parameter is also not appropriate. Mice can be wake without moving.

Indeed, locomotor activity can importantly deviate from time-spent-awake in certain situations. This was however not the case in the current experiments as could be deduced from the very strong positive correlation between waking and LMA (R^2^=0.79) which we presented in Suppl. Figure 5A (now Suppl. Figure 6B), confirming that the two variables strongly covary. Moreover, from a modeling perspective the important point is not whether locomotor activity one-to-one relates to waking but that the time courses of the two variables match. We have added another panel to this supplementary figure illustrating the time course of LMA and waking to underscore their similarity (Suppl. Figure 6A). Finally, we now have modelled PER2 dynamics in the 6h SD experiments based on either wakefulness or LMA. Both resulted in similar tight fits. These findings were added to the results and Suppl. Figure 6C. To conclude, in the context of the model, locomotor activity is a highly appropriate proxy for the sleep-wake distribution.

5. In the method section, the authors described each procedure. However, it was impossible for me to know how exactly each individual mouse went through the procedures. It seems some sleep deprivation was done under dim red light and other was done under light. I suggest the authors provide a timeline (e.g. flowchart figure) of procedures in each experiment.

We thank the reviewer for this useful suggestion and we have included the suggested figure as Suppl. Figure 1.

6. Since in vivo bioluminescence recording became popular technology, I strongly suggest the authors move the data presented in supplementary Figure 2 to the main text as an actual figure. Were CAG mice also crossed with hairless strain? Does tissue explanted from this mouse in vitro not show a circadian rhythm? It even looks like luminescence from the mice with luciferin supplemented by osmic pump shows a weak rhythm. I wonder if CAG promoter may have circadian rhythm as CMV promoter does (Collaco et al., 2005). Amplitude of activity rhythm is lowered in mice with pump. Is this due to surgery (pump implantation)? If so, lowered amplitude of luminescence rhythm might also be affected by surgery. Having a control (saline in pump and luciferin supplied through drinking water) is necessary.

Although the many suggested follow-up experiments related to the results obtained in the CAGmice have their merit, addressing them experimentally is clearly far removed from the aims of our manuscript and would be more appropriate for a separate, technical paper. With a straightforward and unique approach, we show that administering the luciferase substrate in the drinking water gives rise to high-amplitude bioluminescence rhythms that all but disappear when using mini-pumps. The by far most plausible explanation is that the large-amplitude rhythms are an artefact associated with the circadian timing of water intake (and thus luciferin availability) and that therefore mini-pumps are to be preferred to administer the substrate.

Indeed, a small yet significant bioluminescence rhythm is present in CAG mice implanted with mini-pumps which does, however, not challenge our choice of route of administration. Moreover, this residual bioluminescence rhythm peaks 6.3 circadian hours prior to when the activity rhythm peaks and bioluminescence thus decreases when animals are awake most, again demonstrating that changes in bioluminescence do not simply follow changes in e.g. body temperature associated with the circadian sleep-wake distribution.

The reviewer claims that the amplitude of the activity rhythm in mice implanted with minipumps is lower. Statistics fail to support the reviewer’s claim (P=0.67, t=0.47, paired t-test, n=4 on amplitudes derived from sine-wave fits of individual time-course data).

Reviewer #3:This is an interesting study. As far as I know, there are no previous studies performing simultaneous measurement of sleep-wake state and bioluminescence (clock gene expression) in freely moving mice. In addition, their measurement method was well validated through several pilot studies, and measurements obtained using this method are therefore reliable. In fact, the presented data are clear and convincing. However, although I have no doubt that the results are useful and valuable for a wide range of circadian scientists, I feel the present version of their work may not be a strong candidate for publication in this journal because of insufficient novelty and impact.

We thank the reviewer for judging our work well validated and our data reliable, clear, and convincing. Our study is indeed the first to simultaneously record EEG/EMG and bioluminescence, as well as locomotor activity allowing to show that undisturbed sleep-wake behavior contributed to PER2 dynamics.

1) The results clearly revealed the temporal relationship between sleep-wake state and PER2 bioluminescence through simultaneous monitoring in freely moving mice, with high time resolution. However, because previous studies have already found that sleep deprivation results in a change in PER2 levels, I feel that the present study does not make obvious progress toward further understanding the mechanism. The simultaneous monitoring experiments are interesting but each of the techniques was originally established by other groups. For example, some findings related to the identification of systemic or molecular pathways controlling PER2 expression during sleep deprivation may be required.

The reviewer states our work lacks the novelty that would justify publication in *eLife* by referring to previous sleep-deprivation studies. In doing so, the reviewer overlooked an important novel aspect of our current study: we are the first to have quantified the relationship between spontaneous sleep-wake behavior and PER2-bioluminescence, whereas our previous work only assessed the effects of sleep deprivation. We discovered that like sleep deprivation, also spontaneous wakefulness, raises PER2 levels. The finding that PER2’s increase is associated with wakefulness *per se* and does not result from confounding physiological changes accompanying the sleep deprivation (e.g. stress, see next issue), is a core assumption of the mathematical model which, with high accuracy, could predict PER2’s behavior under three challenging experimental protocols. Thus, the circadian distribution of sleep-wake behavior under undisturbed conditions, considered by most a mere output of the clock, modulates the circadian time-keeping circuitry and is required to maintain high-amplitude PER2 oscillations. We believe this to be a major advance in understanding circadian-rhythm generation in tissues peripheral to the SCN as it challenges current dogma of the clockgene circuitry being responsible for all rhythmic gene expression, including clock genes themselves. In the Discussion we had already reviewed the molecular pathways by which sleep-wake state could affect the clock-gene circuitry. We recently published on one of these pathways (temperature – CIRBP – clock genes; Hoekstra *et al. eLife* 2019) through which the sleep-wake distribution could modulate clock-gene expression outside the SCN. We now have put more emphasis on the importance of spontaneous sleep-wake behavior on PER2 bioluminescence rhythms and added an analysis documenting this further (l. 344-353).

2) Although a detailed description of the method for sleep deprivation is important, I was not able to find it in the main text.

We indeed did not describe the sleep-deprivation method in detail but referred to our sleep- phenotyping methods paper instead [“…mice were sleep deprived for six hours at a time they were expected to rest (….) by gentle handling as described (Mang and Franken, 2012)”]. We have expanded on this method in the revised manuscript (l. 683-689).

The authors should clarify whether sleep deprivation itself or other factors such as stress signals triggered by sleep deprivation affect PER2 expression. At this time, I feel not sleep deprivation per se but some stress signals affect PER2 expression. Many previous studies have shown that glucocorticoids, well-known stress hormones, strongly affect peripheral clocks. Sleep deprivation, one of the stimuli activating stress signals, may therefore affect PER2 expression in peripheral clocks via glucocorticoid signaling pathways. The contribution of glucocorticoids to PER2 expression during sleep deprivation can be examined simply through surgical removal of the adrenal glands.

Short-term sleep deprivation by gentle handling is indeed considered a mild stressor and corticosterone does increase as we had already clarified in the Discussion (“How does wakefulness increase PER2?”). We published the experiment the reviewer suggested a while ago (Mongrain *et al.* Sleep 2010). This study shows that the increase in *Per2* expression during sleep deprivation is not solely due to the sleep-deprivation associated corticosterone increase as in adrenalectomized mice sleep deprivation still significantly increases *Per2* levels. These results, together with the results obtained after spontaneous periods of waking (which are not accompanied by an increase in corticosterone) presented in the current study, strongly argue against the simple scenario proposed by the reviewer. These issues are addressed in the Discussion (l. 499-511).

3) Their conclusions may be specific to the kidney. All bioluminescence data were obtained through monitoring only the kidney. The authors should justify why they can conclude that sleep deprivation affects "peripheral clocks" only judging from kidney data. To generalize kidney data to other peripheral clocks, they should repeat similar experiments using at least another organ or tissue. Real-time monitoring of bioluminescence from the liver is technically simple because injected adenoviruses (carrying a reporter gene driven by a clock gene promoter) specifically infect the liver.

In an earlier publication (Curie *et al.* Sleep 2015), using an in vivo imaging approach, we already demonstrated that sleep deprivation increases PER2 in cortex, liver, and kidney, which, for cortex and liver, were confirmed by western blot. The baseline and recovery bioluminescence dynamics obtained in central and peripheral tissue with the RT-Biolumicorder depicted in Figure 1A match remarkably well those reported for cortex and liver and kidney, respectively, with the in vivo imaging method. We are therefore confident that the effects we report can be generalized to other organs peripheral to the SCN.

When setting up the experiments, we also recorded central (cortical) PER2bioluminescence together with EEG/EMG in 6 C57BL/6J PER2::LUC mice. Because detected bioluminescence levels were somewhat noisier (l. 167-169), for the original manuscript we decided to focus on the better accessible (hairless SKH1 mice) and more robust peripheral (kidney) signals. Nevertheless, also the cortical bioluminescence signals demonstrate that the results we reported are not specific to kidney, albeit with tissue-specific dynamics during spontaneous wake episodes. We have included these central data in Figure 1 and addressed the tissue-specificity in the Discussion (l. 531-551).

[Editors’ note: what follows is the authors’ response to the second round of review.]

Reviewer #1:In this paper entitled "The sleep-wake distribution contributes to the peripheral rhythms in PERIOD-2", the authors attempted to find the factor(s) directly contributing to clock gene expression in peripheral tissues. To achieve that, they observed PERIOD-2 (PER2) expression in peripheral tissues of freely moving mice and developed new mathematical models to analyze them. They found that the sleep-wake distribution correlates to the peripheral PER2 dynamics, and that the peripheral PER2 rhythms can be predicted by sleep-wake-state- and peripheral-circadian-dependent factors. Despite the difficulty to dissect the effects of circadian and the sleep-wake cycle on gene expression in vivo, they provided direct evidence that the sleep-wake cycle itself affects peripheral daily rhythms by the unique experimental designs. However, there are still a few points to be addressed before publication including a detailed validation of the PER2 bioluminescence system, which would further support their conclusion.All things considered, their findings will have an impact on the chronobiology field, as giving direct evidence on sleep-wake-state-dependent regulation of peripheral clock gene expression. In addition, those methods can be useful to investigate the gene expression not only in the kidney or cerebral cortex but also in other various tissues of freely moving mice.In the revised paper, the authors addressed almost all our concerns with proper correction and additional analyses, but it has some points to be improved before publication, which are summarized in the comments below.1. The authors assumed that PER2 bioluminescence directly reflected expression of PER2 protein, but it has not been fully confirmed yet especially in the fast (within an hour) dynamics. For further progress of the method, the combination with other techniques to quantify fast and small (< 10%) protein changes would be required, as the authors mentioned in the paper.

We fully agree with the reviewer and if techniques were available that could reliable quantify the small but significant changes in protein levels that we reported on, we would have readily used them. However, for several reasons this is less straightforward as it might seem: (i) Western blots are not sensitive enough to pick up such small differences as this method is semi-quantitative at best and (ii) the design can no longer be longitudinally (i.e. within the same individual) in contrast to our current in vivo *t*echnology. This latter point is important when assessing the effects of spontaneous wakefulness on PER2 as this would require each animal to be closely monitored (using EEG/EMG signals) for periods of either sleep or wake of similar duration to appear at approximately the same circadian phase, before rapidly killing the mouse for tissue extraction such that waking up (if asleep) and stressing the animal does not affect the measurement. Moreover, because of the semi-quantitative nature of Western blots, small effect size, and group instead of within-subject comparisons, we would need to kill a large number of mice for quantifying PER2 for each time point and a time course analyses, as presented here, would be practically impossible. For these reasons, we decided to analyze bioluminescence dynamics in mice with other luciferase reporter constructs to rule out nonspecific sleep-wake related effects on the bioluminescence reaction. See our response to the first comment by Reviewer 2 below.

2. This reviewer suggests that the authors should discuss about the controversial results regarding PER2-bioluminescene amplitude and sleep-wake distribution in Figure 3 and Suppl. Figure 7. When the amplitude of the sleep-wake distribution was reduced, PER2-bioluminescence amplitude was reduced in 4/6 mice as the model expected, while it was increased in 2/6 mice.

We believe that, given the data, we went as far as we could in trying to explain these counterintuitive results. Our model gave us a tool to investigate this issue and the simulation results suggest that the phase of the circadian force relative to the phase of the wake force could be a contributing factor. This scenario was presented and discussed in detail in the Results section (Lines 386-410). However, this explanation follows from an *in silico* analysis and, as we pointed out in that section, it is clear that this hypothesis must be tested by follow-up experiments before making any firmer claims.

3. Regarding Figure 3E, the authors showed that the 2hOnOff protocol decreased the amplitude of sleep-wake distribution. However, the sleep-wake distribution during the recovery phase was not assessed. Thus, the reviewer recommends that the authors should add data of recovery day to Figure 3E and compare it to the PER2 bioluminescence trends (Figure 3C).

We recorded recovery after the 2hOnOff protocol, but did not further analyze the sleep-wake distribution during this period. Following the reviewer’s recommendation, we now have analyzed sleep-wake states for 48h of recovery and added the estimated amplitude of the sleep-wake distribution during these two recovery days to Figure 3E. After the reduction of the daily sleep-wake amplitude during the 48h 2hOnOff experiment, during recovery amplitude was restored to baseline levels.

4. In lines 290 to 294, the authors concluded that the prediction from the model, the behavioral rhythm reduction reduces PER2 bioluminescence amplitude, which is consistent with their observation in the Figure 3. However, this conclusion could be difficult to understand because the 2hOnOff protocol increases the PER2 bioluminescence amplitude in one-third of mice (there may be no significant change between BSL and the 2hOnOff protocol; see Figure 3C). The authors provided an additional model (Suppl. Figure 7A-C) to explain the individual difference, and thus the reviewer suggests the authors to describe that conclusion after the modeling part to make the logic clearer.5. Regarding Suppl. Figure 7A-C, the authors described the increase in PER2 amplitude during the 2hOnOff experiment by advancing the phase of FPERI more than 2.5h. The reviewer suggests that the authors could mention these results in Results or Discussion section.

At the group level the reduction in PER2 amplitude during the 2hOnOff protocol did not reach significance, as was clearly stated in the Results section and figure legend. Yet, when inspecting the individual responses, we found an interesting pattern suggesting that the 2hOnOff protocol did affect PER2 amplitude albeit in opposite direction for individual mice (Lines 251-261). With the analysis presented in Suppl. Figure 7A-C (now Figure 4 —figure supplement 2A-C), we did not introduce an additional model to explain this phenomenon. Instead, we used the same model as proposed earlier, but instead of letting the model determine a single (optimal) phase of the circadian force, we systematically varied it to predict the resulting effects on PER2 amplitude. We described the rational and results of this *in silico* experiment in the Results section after the modeling part (Lines 386-410). To enhance clarity, we have now commented after line 256 (now Lines 260-263) on the results of the modelling section, stating that variation in circadian phase could underlie the differential individual response.

Reviewer #2:Hoekstra et al. investigated the relationship between sleep state and PER2::LUC luminescence in freely behaving mice and showed that luminescence acutely changed at sleep/wake transitions. The authors also showed that daily sleep deprivation enhanced the PER2::LUC luminescence rhythm amplitude in SCN ablated mice.A strength of this study is the simultaneous recording of EEG/EMG/activity and PER2::LUC luminescence in freely moving mice.Weakness: One obvious concern is that acute PER2::LUC luminescence changes occurring at sleep/wake transitions are artifacts (e.g. due to acute change in core body temperature), but not caused by synthesis/degradation of PER2 protein. Because those luminescence changes are relatively large and fast (30-40% change within 30 min), a conventional method may be used for validation. The authors recorded activity and bioluminescence from non-circadian luminescence reporter mice (CAG -luc). Alternatively, the authors can analyze that data and exclude artifact possibility if luminescence change doesn't occur at active/inactive transitions.

Thank you for this great suggestion. With the reviewer we would have preferred to use a conventional method for confirmation but as explained in our answer to the first comment of Reviewer 1 this is unfortunately less straightforward as it might seem.

Following your suggestion, we analyzed bioluminescence dynamics at the onset of activity bouts in the CAG-Luc mice. As you can see in and the new Figure 1 —figure supplement 5, activity bouts in CAG-Luc mice are accompanied by robust increases in bioluminescence, even larger in amplitude than those observed in *PER2^Luc^* mice. This result is consistent with the CMV major-immediate early enhancer/promotor, which is part of the CAG synthetic promotor driving luciferase, being rapidly activated by large number of transcription factors such as NF-κB, CREB/ATF, and AP-1 (see review by Stinski and Isomura Med Microbiol Immunol 2008; doi: 10.1007/s00430-007-0069-7), a response comparable to that of an immediate early response such as c-FOS (see e.g. Brightwell *et al.* Gene 1997; doi: 10.1016/S0378-1119(97)00178-9 and Collaco and Geusz BMC Physiol. 2003; doi: 10.1186/1472-6793-3-8).

We used the CAG-Luc mice to verify the effect of the route of luciferin administration on circadian bioluminescence dynamics because we expected this synthetic promotor not to be regulated in a circadian fashion (which we confirmed). We did, however, not chose this mouse line to rule out that PER2bioluminescence changes at sleep-wake transitions are artefacts, an issue that was raised during the review process. After discussing this issue with Prof. David Gatfield, a colleague at my institute studying circadian aspects of translation, he shared some of his recordings with other luciferase-reporter constructs using the same set-up (i.e., osmotic mini-pumps delivering luciferin and the RT-biolumicorder system recording bioluminescence). We recorded bioluminescence in mice expressing luciferase under the control of the promotor of the housekeeping gene *Pgk1* (*Pkg1-Luc* mice). As expected of a housekeeping gene, overall bioluminescence levels do not consistently vary during the recording and no activity-bout related changes were observed (Figure 1 —figure supplement 5). These recordings demonstrate that the increases in PER2 bioluminescence we observed at sleep-wake transitions are not an artifact of the sleep-wake transition as similar activity-related increases should have been observed for all 3 constructs. We have added the *Pgk1-Luc* results together with the CAG-Luc results to the Results section (Lines 177-182) and added a new supplementary figure (Figure 1 —figure supplement 5). For details on the *Pgk1-Luc* mice see Methods.

Another weakness is the PER2::LUC luminescence measurement from the brain. The authors must show IVIS imaging and show the signal coming from the brain and not from other tissues.

We performed the experiment the reviewer suggested in a previous publication using IVIS imaging (Curie *et al.* Sleep; 2015 doi: 10.5665/sleep.4974). As described in the Results section these experiments were performed in C57BL6/J mice with pigmented pelage absorbing emitted photos. In addition, the substrate luciferin was administrated only centrally and the by far largest (i.e., only) source of photons in this setup is the glass cone directly mounted on top of the skull over the left cortical hemisphere. Moreover, in the same publication in situ immune-histofluorescence indicated the cerebral cortex as the main source of the almost 2-fold increase in brain bioluminescence after a 6h sleep deprivation. This work was cited when we first stated that brain signal represents cortical activity but now have stated this more explicit (Lines 126-130).

Abstract: Line 18, 19: instead of saying PER2 increases or PER2 rhythmicity, I suggest using PER2::LUC luminescence increases, PER2::LUC luminescence rhythmicity.

We changed the abstract to “PER2 bioluminescence” increases and rhythmicity, consistent with the terminology used in the rest of the manuscript.

Line 89, Suppl. Figure 1: should be Suppl. Figure 3.

We intended referring to the protocol figure (previously Suppl. Figure 1) here but now have added the reference to the other figure as suggested.

Figure 1B: The hypnogram only includes sleeping and waking, please include REM and NREM. I understand that the authors only wanted to indicate sleep/wake transition used for the luminescence analysis indicated in C, but I think the authors should plot standard hypnogram.

As the reviewer noted, this is not a classical hypnogram but our way of indicating which sleep-wake and wake-sleep transitions met our criteria and contributed to the analyses. In the manuscript we made no statements on the two substates of sleep (REM sleep and NREM sleep), and their respective possible effects on PER2 bioluminescence. Moreover, adding separate tracks for REM sleep and NREM sleep will clutter the figure and might confuse the reader. We therefore prefer to keep the figure as is but we now give values of REM sleep and NREM sleep obtained in the recordings on which Figure 1B is based in the text (Lines 122-125) to assure the reader that NREM and REM sleep were expressed at expected quantities under the different recording conditions.

Figure 1C: To avoid confusion, it is better to use the same colors as in A and B (pink: peripheral, purple: central).

In this figure panel we now have used the same red and purple hues for peripheral and central transition data, respectively.

Figure 2: The authors should show individual records in the supplemental information. I know the authors discussed two possibilities that can explain changes in amplitude in group average data. The amplitude of an individual mouse was shown in Figure 2C. But I think, like me, the readers will still want to see the individual record. The amplitude of 3 cycles in REC should be analyzed separately (Plot as REC day 1, REC day 2 and REC day 3). Technically REC day 1 is the same as SD, but it is good to show how the amplitude in the group average and the individual mouse declines.

We have added these data to Figure 2 —figure supplement 2.

Line 573-584: This is an important discussion addressing the possible temperature artifact. However, all statements are based on analysis, "data not shown", unfortunately. The authors must show the data.

We have added these data to the new Figure 1 —figure supplement 6.

Line 763: I think the activity data was binned, not averaged.

We indeed binned the data into (circadian) hourly intervals. However, values in corresponding circadian hours were then averaged across individuals. We have now specified this (see Lines 818-820).

Supplementary Figure 1: I know luminescence recording was done in DD, but please indicate this, as the lighting condition in the other treatments was stated.

We have added this information to the figure.

Supplementary Figure 3B: Please indicate individual records as well, so the relationship between luminescence and activity can be evaluated.

We have now provided an example of individual recordings of the CAG-Luc bioluminescence in Figure 1 —figure supplement 5, as well as an assessment of the association between changes in activity and bioluminescence in the CAG-Luc mice.

Reviewer #3:Response to the comment #1It is my understanding that almost all of the conclusions made by the authors are based on the temporal correlation between kidney-derived bioluminescence and sleep-wake states. Therefore, I do not feel convinced about these conclusions without additional experimental evidence suggesting the causal link between spontaneous wakefulness and PER2 expression.Response to the comment #3The authors claim that spontaneous wakefulness-driven PER2 increase is one of the major findings. Therefore, under not sleep deprivation but spontaneous wakefulness conditions, they need to confirm PER2 expression in other organs and apply expression data to their mathematical model.

We had already included brain (cortex) derived PER2-bioluminescence dynamics at sleep-wake transitions in our revised manuscript submitted in May 2021 (see Figure 1). As similar dynamics were observed at sleep-wake transitions in both cortex and kidney, we demonstrate that our observations go beyond the kidney and also beyond a mere correlation as PER2 bioluminescence consistently follows spontaneous as well as experimentally induced wakefulness in both tissues. Moreover, we were able to predict the brain dynamics of PER2 with the model parameters set for kidney (Lines 329-331), confirming that at least mathematically sleep-wake driven PER2 dynamics in kidney and cortex underlie similar (but not identical) rules. See next our reply concerning the phase of the SCN.

A few previous studies, although not cited in the present version, clearly demonstrated that arousal levels affect the circadian clock via the SCN (for example, PNAS 2016 Yamakawa el al). I think that the present data may not make obvious progress from these previous achievements.

The reference mentioned does not take position on clock-gene rhythms because their expression was not quantified. Our work highlights the importance of the sleep-wake distribution when evaluating clock-gene rhythms in tissues peripheral to the SCN (including the cortex) in the mouse. Moreover, altered ‘arousal levels’ such as associated with a sleep deprivation (Vassalli and Franken, PNAS 2017; doi: 10.1073/pnas.1700983114) or with food-restriction conditions, which redistributes activity patterns (Damiola *et al.*, Gen Dev 2000; doi: 10.1101/gad.183500), does not affect clock-gene rhythms in the SCN in the mouse (see Discussion Lines 520-521). Therefore, at least under entrained conditions, the phase of clock-gene rhythms in the SCN remains locked to the light-dark cycle, while phase in tissues peripheral to the SCN (including the cortex) can be dissociated from that in SCN by e.g. an altered sleep-wake distribution.

Many previous studies have indicated that circadian phase in clock gene expression varies among peripheral organs and tissues, and it is therefore difficult to generalize the temporal correlation between single organ-derived bioluminescence and sleep-wake states to other peripheral and central clocks. In particular, circadian phase of the SCN is largely phase-advanced in comparison with that of peripheral clocks, raising a question on the author's thought that spontaneous wakefulness is a similar trigger of PER2 in both peripheral and central clocks.

[Editors’ note: what follows is the authors’ response to the third round of review.]

There are only two final pending issues that we anticipate should be easy to incorporate as part of the results/discussion and be included in the final rebuttal letter, without the need for any additional experiments.Essential revisions:1) Although the new data has clearly enriched the paper, this still raises the concern of one of the reviewers regarding the presence of low amplitude baseline rhythm in luminescence (particularly as CAG-luc appears to be activated by wake: the reviewer expects to see a basal circadian change of luminescence in CAG-luc mice).

This might indeed be expected provided that the CMV activation upon waking up (i) does not vary with circadian time and (ii) does not again decreases as wakefulness progresses. From the example in Figure 1 —figure supplement 5 it seems that the wake-related activation is strongest after consolidated bouts of sleep (i.e. rest phase), and weakest during the active phase, when the fewer and shorter periods of sleep do not appreciatively decrease the CAG-luc signal. Therefore, although CAG-luc bioluminescence increases after sleep-wake transitions, a circadian modulation in this response might explain the lack of a robust modulation of the average hourly bioluminescence levels. If examining the precise dynamics linking the bioluminescence signal of this synthetic promotor to the sleep-wake distribution is considered of interest by the scientific community, this should be addressed experimentally – however this clearly falls outside the scope of our PER2 work. We only invested in recording CAG-luc mice to assess which route of luciferin administration (drinking water *versus* osmotic mini-pumps) was best.

2) Without showing an absolute luminescence value emitted from the brain, liver, and kidney in the same mouse, it cannot be utterly said that the luminescence signal from the entire mouse is just coming from the cortex. Part of this concern is based on the fact that the Curie et al., 2015 work not only shows signals (in B6 mice) coming from the cortex, but also clear signals coming from the kidney/liver (as visualized with the IVIS signals, whereas the RT-Biolumicorder does not allow sorting those signals).

In the Curie *et al.* 2015 publication we indeed recorded bioluminescence in the brain, liver, and kidney of B6 mice. However, recording of peripheral and central signals has to be done in separate animals for two reasons. First, where luciferin is delivered (central *versus* peripheral) is key to where bioluminescence is detected, as luciferin does not readily cross the blood-brain barrier (Bakhsheshian *et al.* PNAS 2013; PMID: 24297888). To further facilitate central detection of bioluminescence, a glass cone was directly mounted on a thinned skull allowing passage of photons generated by the underlying cortex. Second, the body of the B6 mice, the strain we used specifically to detect central PER2 bioluminescence, was not shaved, thereby preventing the release of photons. In B6 mice thus prepared (not shaved, a cannula to infuse luciferin centrally, and a glass cone placed over the cortex), the PER2-bioluminescence signal by and large represents photons emitted by the brain (see Figure 1—figure supplement 2). The tissue-specific PER2-bioluminescence dynamics observed after sleep deprivation, which were confirmed by western blot analyses and brain immune-histofluorescence, confirm that the bioluminescence originated from different sources. We now have further emphasized in the Results section the specific preparation of the animals used to record central PER2 bioluminescence (Lines 102-106), detailed the IVIS imaging experiment in the Methods (Lines: 681-686), and added to Figure 1 —figure supplement 2 the image of central bioluminescence detected with the IVIS setup, clearly showing that besides the head no other above-background signal could be detected from the animal.